

# Excess methane, ethane, and propane production in Greenland ice core samples and a first isotopic characterization of excess methane

Michaela, Mühl[1], Jochen Schmitt[1], Barbara Seth[1], James E. Lee[2], Jon S. Edwards[3], Edward J. Brook[3], Thomas Blunier[4], Hubertus Fischer[1]

[1]Climate and Environmental Physics and Oeschger Centre for Climate Change Research, University of Bern, Bern, 3012, Switzerland
[2]Los Alamos National Laboratory, Earth Systems Observation, Los Alamos, NM 87545, USA
[3]College of Earth, Ocean, and Atmospheric Sciences, Oregon State University, Corvallis, OR 97331, USA
[4]Centre for Ice and Climate, Niels Bohr Institute, University of Copenhagen, Copenhagen, 2200, Denmark

*Correspondence to*: Michaela Mühl (michaela.muehl@unibe.ch)

**Abstract.** Air trapped in polar ice provides unique records of the past atmospheric composition ranging from key greenhouse gases such as methane ($CH_4$) to short-lived trace gases like ethane ($C_2H_6$) and propane ($C_3H_8$). Provided that the analyzed species concentrations and their isotopic fingerprints accurately reflect the past atmospheric composition, biogeochemical cycles can be reconstructed. Recently, the comparison of $CH_4$ records obtained using different extraction methods revealed disagreements in the $CH_4$ concentration for the last glacial in Greenland ice. Elevated methane levels were detected in dust-rich ice core sections measured discretely pointing to a process sensitive to the melt extraction technique. To shed light on the underlying mechanism, we performed targeted experiments and analyzed samples for methane and other short-chain alkanes ethane and propane covering the time interval from 12 to 42 kyears. Here, we report our findings of these elevated alkane concentrations occurring in dust-rich sections of Greenland ice cores. The alkane production happens during the melt extraction step (*in extractu*) of the classic wet extraction technique and reaches 14 to 91 ppb for $CH_4$ excess in dusty ice samples. We document for the first time a co-production of excess methane, ethane, and propane (excess alkanes) with the observed concentrations for ethane and propane exceeding their past atmospheric background at least by a factor of 10. Independent of the produced amounts, excess alkanes were produced in a fixed molar ratio of approximately 14:2:1, indicating a shared origin. The amount of excess alkanes scales linearly with the amount of mineral dust within the ice samples. The isotopic characterization of excess $CH_4$ reveals a relatively heavy carbon isotopic signature of (-46.4 ± 2.4) ‰ and a light deuterium isotopic signature of (-326 ± 57) ‰ in the samples analyzed. With the co-production ratios of excess alkanes and the isotopic composition of excess methane we established a fingerprint that allows us to confine potential formation processes. This fingerprint is not in line with a microbial

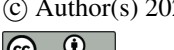



origin, rather such an alkane pattern is indicative of abiotic decomposition of organic matter as
found in sediments, soils and plant leaves. This study provides first indications for an abiotic
reaction producing excess alkanes during ice core analyses and discusses potential mechanisms.
We see an urgent need to correct the already existing discrete $CH_4$ records for excess $CH_4$
contribution ($CH_{4(xs)}$, $\delta^{13}C$-$CH_{4(xs)}$, $\delta D$-$CH_{4(xs)}$) in dust-rich intervals in Greenland ice.
Specifically, excess $CH_4$ has a significant effect on the assessments of the hemispheric $CH_4$
source distribution. As we observe that in some intervals excess $CH_4$ is in the same range as the
Inter-Polar Difference, previous interpretations of relative contribution of high latitude northern
hemispheric $CH_4$ sources need to be revised.
**1. Introduction**
Atmospheric air entrapped in polar ice represents a unique archive of the past atmospheric
composition including the concentration of greenhouse gases like carbon dioxide ($CO_2$)
methane ($CH_4$) and nitrous oxide ($N_2O$) but also short-lived trace gases such as ethane ($C_2H_6$)
and propane ($C_3H_8$). The ongoing anthropogenic increase in the atmospheric concentrations of
these gases and the global warming caused by it makes a detailed understanding of their
preindustrial variations and biogeochemical cycling of paramount importance and only polar
ice cores are able to provide this information. However, to interpret reconstructions of the
atmospheric composition from polar ice cores requires that archived atmospheric trace gases
are not altered within the ice itself. Furthermore, the air must be extracted from the ice sample
without altering the original composition. Thus, the comparison of ice core records obtained
using different extraction techniques and from different ice cores requires careful consideration
and interpretation.

It is known that not all drill sites or specific time intervals are equally suitable to derive pristine
atmospheric trace gas records, for example $CO_2$ data from Greenland ice are subject to $CO_2$ in
situ production due to impurities in the ice (Anklin et al., 1995; Smith et al., 1997). In situ
production is also observed for $N_2O$, for example in glacial Antarctic ice core samples
characterized by higher dust content (Schilt et al., 2010). In contrast, $CH_4$ in polar ice cores has
been traditionally interpreted as "the good guy", which in the absence of melt layers is not
affected by such processes. However, more recent results from Greenland showing elevated





CH$_4$ concentrations in glacial dusty ice (Lee et al., 2020) and high amplitude CH$_4$ spikes in
Holocene ice (Rhodes et al., 2013, 2016) question this assumption.
This becomes especially worrisome as atmospheric methane also shows a North-South gradient
reflecting the predominance of Northern Hemisphere sources. Bipolar ice core studies have
been used to quantify this Inter-Polar Difference (IPD) in past CH$_4$ concentrations (Chappellaz
et al., 1997; Baumgartner et al., 2012, Beck et al., 2018) with the goal to derive the contribution
of northern and southern hemispheric sources to the overall CH$_4$ changes. The Holocene IPD
is on the order of several tens of ppb, i.e., one order of magnitude smaller than the   past
atmospheric CH$_4$ concentration. Thus, any small CH$_4$ bias on the order of a few ppb has a strong
impact on the conclusions drawn from this IPD, while the error on the total radiative forcing by
such small biases is negligible. In summary, existing results of CH$_4$ concentrations from
Greenland and Antarctic ice cores have to be carefully scrutinized for such effects.
A first step in this direction has been made in previous work by Lee et al. (2020), for example
by comparing CH$_4$ records derived using different measurement techniques. Past CH$_4$
concentrations ([CH$_4$]) are retrieved by measurements of Greenland and Antarctic ice cores
using traditional discrete and relatively new continuous melt extraction techniques. While
discrete ice measurements deliver one single value for each sample, Continuous Flow Analyses
(CFA) gradually melt a thin prismatic stick of the ice core providing a continuous record for
this section. Although in both techniques the ice sample is melted, the CFA technique separates
air from the melt water stream in about 1-2 min providing only a short time for any reaction in
the water while for the discrete technique the contact time is typically 15-30 min.

Comparing [CH$_4$] histories from several Greenland ice cores measured discretely (NGRIP,
GISP2, GRIP) with the continuous Greenland NEEM and the continuous Antarctic WAIS
record over the last glacial period, discrepancies in [CH$_4$] between the existing records can be
found in specific time intervals (Lee et al., 2020; Fig. 1 therein). These differences are
particularly visible ~500 years prior to the onset of Dansgaard-Oeschger (DO) event 8 and 12
at around 39.5–40.0 kyears and 48.0–48.5 kyears, respectively, where the discrete NGRIP
[CH$_4$] record shows elevated values (~30 ppb) while the continuous NEEM and WAIS [CH$_4$]
records stay basically flat. Similar observations were also made on the GISP2 and GRIP record
(Lee et al., 2020).
A closer look by Lee et al. (2020) into the existing records revealed further corollaries with
other ice core parameters: intervals with elevated [CH$_4$] in the discrete Greenland CH$_4$ record
correspond to stadial ice with a high abundance of mineral dust (indicated by high Ca$^{2+}$



concentrations), especially visible again prior to DO-8 (and DO-12) when [CH$_4$] and [Ca$^{2+}$]
simultaneously rise. When Ca$^{2+}$ decreases again to low interstadial levels, [CH$_4$] drops by 10-
20 ppb. Note that over the same intervals the corresponding continuous NEEM and WAIS CH$_4$
records remain stable.

Looking at the NGRIP  methane hydrogen isotope ($\delta$D-CH$_4$) record (Bock et al., 2010b) – as
well measured with a discrete melt-extraction technique (Bock et al., 2010a) – it turns out that
in these anomalous sections, as explained above, the isotopic values are also affected. Several
negative hydrogen isotopic excursions with a maximum depletion of 16 ‰ (permil)  prior to
the onset of DO-8 were identified (Bock et al., 2010b). At the time of that publication there was
no straightforward explanation for these depletions (for example by a change in the source
types) that could lead to "lighter" $\delta$D-CH$_4$ values during times of a relatively stable climate.
Using ice from Antarctica much smaller $\delta$D-CH$_4$ variations during this interval were found
(Iseli, 2019), again questioning the atmospheric origin of these $\delta$D-CH$_4$ depletions prior to the
DO onset.

All these variations recorded in Greenland ice give reason to assume that a hitherto unknown
process exists that produces or releases additional methane in some time intervals in Greenland
ice cores (from here referred to as "excess methane" or CH$_{4(xs)}$). This process is related to the
extraction technique (only found in records obtained by discrete melt extractions) and has only
been observed in glacial Greenland ice with high mineral dust concentrations.

A first attempt to characterize CH$_{4(xs)}$ was made by Lee et al. (2020) who analyzed [CH$_4$] in
discrete ice samples with different impurity composition and concentration from several ice
cores (GISP2, NEEM, WAIS, SPICE) using a multiple melt-refreeze technique. With their data
they were able to quantify CH$_{4(xs)}$ contributions of up to 30-40 ppb for Greenland samples.
Sequential melt-refreeze extractions showed that the process leading to CH$_{4(xs)}$ is slow and not
completed during the first cycle (i.e., within around 30 min). A special set of samples was
analyzed with the admixture of a HgCl$_2$ solution to suppress microbial activity in the melt water.
No difference in the measured [CH$_4$] was observed between the poisoned samples and replicates
without HgCl$_2$. In addition, Lee et al., (2020) used the NGRIP [CH$_4$] (Baumgartner et al., 2014)
and $\delta$D-CH$_4$ records (Bock et al., 2010b) to estimate the deuterium isotopic signature of the
CH$_{4(xs)}$. Assuming a two-component mixture of atmospheric methane and excess methane their
model led to a best estimate of (-293 ± 31) ‰ for $\delta$D- CH$_{4(xs)}$.



A straightforward explanation for $CH_{4(xs)}$ may be that $CH_4$ is either produced in the melt water,
or it was produced beforehand and only released during the melt extraction. With respect to
that, Lee et al. (2020) reviewed several mechanisms that could account for the observed
variations in Greenland ice core records. None perfectly matched all their observations but
lastly, three of the proposed mechanisms were short-listed: (1) an adsorption process on dust
particles prior to the deposition on the ice sheet; (2) an in situ production in the ice; or (3) an
abiotic reaction during melt extraction.

Here we resume the work by Lee et al. (2020) and shed more light upon the potential formation
processes using a targeted and more comprehensive study to quantify $CH_{4(xs)}$. We analyzed
specific NGRIP and GRIP ice core samples discretely with two different wet extraction
systems. With our $\delta^{13}C$-$CH_4$ device we are able to measure [methane], [ethane], [propane], and
$\delta^{13}C$-$CH_4$ on a single ice sample in two subsequent extractions. With our second device we add
further data on $\delta D$-$CH_4$. In Sect. 2 we provide information on our sampling strategy and
measurement techniques. With our new experimental results, presented in Sect. 3, we provide
quantitative data for $CH_{4(xs)}$ in NGRIP and GRIP samples and extend our observations to other
"excess alkanes" (ethane and propane), which are revealed to be co-produced during the excess
$CH_4$ production. The observed molar ratios between methane, ethane, and propane are
evaluated and their relation to the abundance of mineral dust ($Ca^{2+}$) within the ice samples is
quantified. A $2^{nd}$ extraction of the melt water enables us to estimate the temporal dynamics of
excess alkane production. Using a Keeling-plot approach to our isotopic results, we calculate
the carbon and deuterium isotopic signature of excess $CH_4$ ($\delta^{13}C$-$CH_{4(xs)}$ and $\delta D$-$CH_{4(xs)}$). Based
on our new and improved evidences, we finally come back to the discussion of the hypotheses
by Lee et al. (2020) in Sect. 4 and offer potential mechanisms that could explain the excess
alkanes in ice core samples.


**2. Ice core samples and measurements**
**2.1 Ice core samples**

Mixing ratios of alkanes (methane, ethane, and propane) and the stable carbon ($\delta^{13}C$-$CH_4$) and
hydrogen ($\delta D$-$CH_4$) isotope ratio of methane were measured on ice core samples from the North
Greenland Ice Core Project (NGRIP) ice core. For this study, a total of 19 NGRIP ice core
samples were measured for $\delta^{13}C$-$CH_4$ and alkane concentrations and nine NGRIP ice samples

 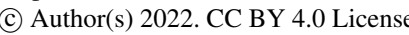

171 for $\delta$D-CH$_4$ covering the depth between 1795.84 m and 1933.25 m. The NGRIP samples stem

172 from the late glacial Marine Isotope stages 3 and 2. These time intervals are characterized by

173 sharp atmospheric CH$_4$ increases in parallel to rapid warmings, the so-called Dansgaard-

174 Oeschger events, but we mostly sampled intervals with stable CH$_4$ concentrations.

175 From the same time period, we also investigate measurements of 41 NGRIP and 12 GRIP ice

176 core samples which were carried out in 2011 and 2018, respectively, and which have not

177 previously been published. See Fig. 1 for an overview of all analyzed NGRIP and GRIP ice

178 core samples over time.

179

180 We also included 22 ice core samples from the European Project for Ice Coring in Antarctica

181 (EPICA) ice core from Dome C that we use as long-term monitoring ice for the system

182 performance and to quantify the blank contribution of the analytical system. Note that Antarctic

183 ice core samples have not shown any signs of CH$_{4(xs)}$.

184 The late glacial time period, which includes the age of most of the measured NGRIP samples,

185 is characterized by an overall high impurity and dust content and low atmospheric methane

186 concentrations. For our analysis, we have selected ice core bags (where for NGRIP and GRIP

187 ice cores, a bag is a 55 cm long ice core section) in which we expect the same atmospheric CH$_4$

188 concentration but a high range of mineral dust content (Ca$^{2+}$). In this way we can compare

189 neighbouring samples that have the same low stadial CH$_4$ levels due to stable atmospheric

190 concentrations and temporal smoothing by firn processes but are expected to vary in measured

191 concentrations due to contributions of excess alkanes. Mineral dust content across our NGRIP

192 samples range from 307 ng/g to 1311 ng/g.

193 This sample selection is also critical to quantify the isotopic signature of the CH$_{4(xs)}$ produced

194 using the Keeling-plot approach (Keeling, 1958). The underlying assumptions of this mass

195 balance approach are that (1) there is only a two-component mixture (atmospheric methane and

196 excess methane) and that (2) the isotopic ratio of the mixture changes by a varying input of the

197 second source (CH$_{4(xs)}$).

198

199 To select the samples, we use high-resolution mineral dust records measured using an Abakus

200 laser attenuation device (Klotz, Germany) for particulate dust (Ruth et al., 2003) as well as Ca$^{2+}$

201 concentrations derived from the Bern Continuous Flow Analysis System (Kaufmann et al.,

202 2008) as dissolved mineral dust tracer (Erhardt et al., 2022). In principle, particulate dust and

203 the specific soluble dust tracer Ca$^{2+}$ are strongly correlated. However, dependent on acidity of





the ice (mainly due to $H_2SO_4$ and $HNO_3$), variable amounts of $CaCO_3$ are converted into soluble
$CaSO_4$ and $Ca(NO_3)_2$ leading to a higher $Ca^{2+}$/ dust ratio (Legrand and Delmas, 1988).
As an example, Fig. 2 shows the $Ca^{2+}$ and mineral dust concentration on the NGRIP depth of
the NGRIP bag 3292 which we used to select the individual samples, and the relevant
parameters measured for each sample of this bag. The data overview for all other measured
NGRIP bags can be found in the Appendix A.
Note that all regression lines are calculated by following the method of York (1968) and York
et al. (2004).

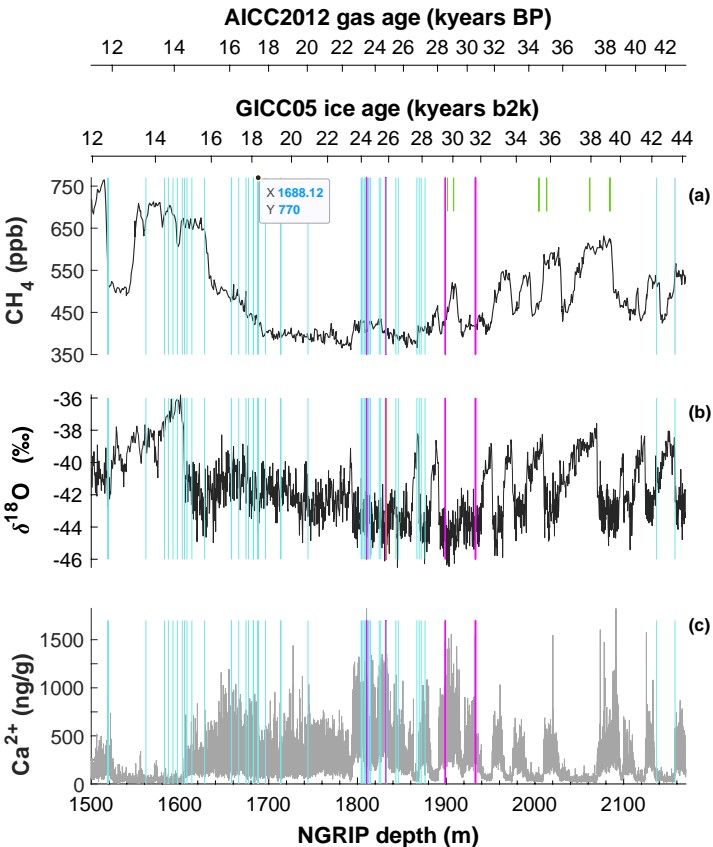


Figure 1: **Overview of the analyzed NGRIP and GRIP samples over time.** All analyzed NGRIP and GRIP ice
core samples are indicated on the NGRIP depth (m) on the bottom axis and the AICC2012 gas age (kyears BP) &
GICC05 ice age (kyears b2k) scale on the upper axes. NGRIP samples measured from the five main bags (3292,
3331 & 3332, 3453, 3515) for the Keeling-plot approach are indicated with vertical lines in pink, NGRIP samples
measured in 2011 and individual NGRIP ice core samples measured in 2019-2020 (not included in the Keeling-
plot analyses) in cyan, and GRIP ice core samples in green. **(a)** [$CH_4$] record measured from NGRIP samples from
Baumgartner et al. (2012, 2014). **(b)** $\delta^{18}O$ record from North Greenland Ice Core Project members (2004). **(c)** $Ca^{2+}$
record from Erhardt et al. (2022).

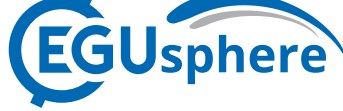



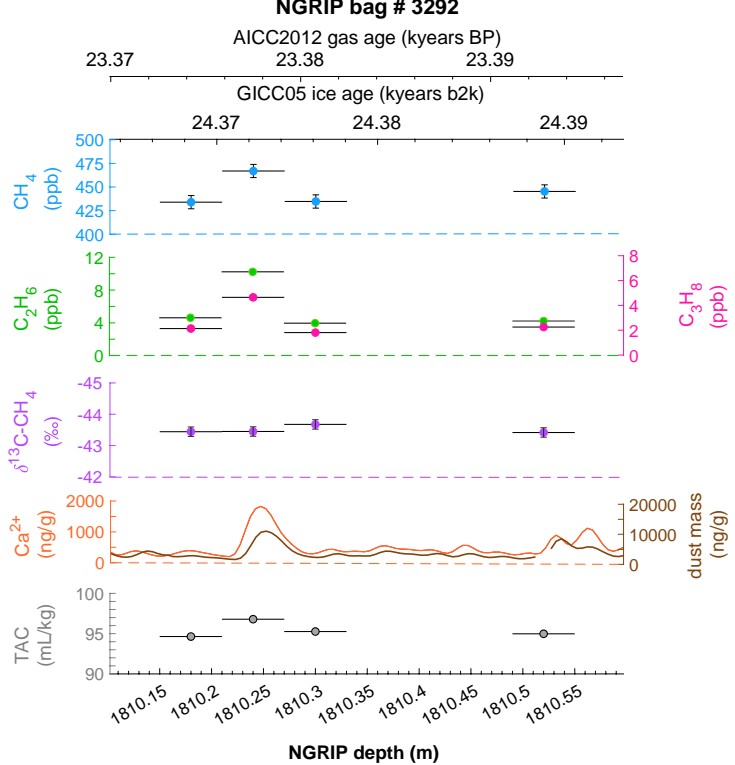

Figure 2: **Detailed data overview for NGRIP bag 3292.** Bag-specific overview of several parameters measured for each sample in this bag: methane, ethane, propane, $Ca^{2+}$, mineral dust mass, TAC (Total Air Content), $\delta^{13}C$-$CH_4$, indicated at the NGRIP depth (bottom axis) and the AICC2012 gas age (upper top axis) and the GICC05 ice age (lower top axis). The mineral dust record is taken from Ruth et al. (2003), the $Ca^{2+}$ record from Erhardt et al. (2022). Here, this is shown exemplarily for the NGRIP bag 3292, the data overview for all further measured NGRIP bags can be found in the Appendix A.



**2.2 CH$_4$, C$_2$H$_6$, C$_3$H$_8$ and $\delta^{13}$C-CH$_4$ Analysis of Ice Core Samples**

The short-chain alkanes and $\delta^{13}$C-CH$_4$ were measured at the University of Bern using the
discrete wet extraction technique as described in detail in Schmitt et al. (2014). With this
method it is possible to measure mixing ratios of methane, ethane, and propane as well as the
methane carbon isotopic signature and other trace gases on a single ice core sample of about
150 g.
Briefly, ice core samples are placed in a glass vessel locked by a stainless-steel flange which is
attached to the vacuum line to evacuate laboratory air (see Fig. 3, step a). Before melting the
ice sample, the leak tightness of the vacuum extraction line is tested with a so-called He blank.
The ice sample is then melted under vacuum with the help of infrared radiation for ~35 min to
release the enclosed air (step b). The released air is continuously removed from the sample
vessel by a pressure gradient towards an adsorbing AirTrap (activated carbon), collecting all
relevant air components at -180°C. After melting is completed, the temperature of the melt
water is stabilized close to 0°C. Afterwards, He is flushed for ~14 min through a capillary at
the bottom of the vessel to bubble He through the melt water to transfer any remnant gas species
dissolved in the melt water onto the AirTrap (step c). The sample vessel is then sealed by closing
inlet and outlet valves (step d). Consecutively, the AirTrap is warmed up in two steps to first
remove N$_2$ and O$_2$ and in a second step to release the gases of interest which are then sent after
a cryofocus step to the gas chromatograph (GC) for separation and quantification using an
isotope ratio mass spectrometer (Isoprime 100, Elementar).

Precision of this method for CH$_4$ is about 5 ppb, 0.15 ‰ for $\delta^{13}$C-CH$_4$, and for both C$_2$H$_6$ and
C$_3$H$_8$ the precision is 0.2 ppb or 5 % (whatever is higher) (Schmitt et al., 2014) for the typical
NGRIP samples used in our study, where isotopic data are expressed using the $\delta$ notation on
the international Vienna Pee Dee Belemnite (VPDB) scale. Blank levels for these species using
this device are at 4 ppb for CH$_4$, 0.4 ppb for C$_2$H$_6$ and 0.3 ppb for C$_3$H$_8$.

With their experimental investigations, Lee et al. (2020) were already able to demonstrate that
production/ release of CH$_{4(xs)}$ is time dependent. We therefore conclude that this process does
not have to be completed in the time available for the gas extraction described above. We
continued the analyses of excess alkane production with an additional extraction step (here
referred to as 2$^{nd}$ extraction, steps d-g in Fig. 3) following the normal ice extraction routine.
After all sample air is collected in the 1$^{st}$ extraction, the melt water is left in the sample vessel
and held at temperatures close to 0°C for ~100 min (step d). After this "waiting time" of ~100



min, He is purged through the melt water for ~24 min to extract the gases that have been
accumulated during this time interval (step f). The gases from this 2nd extraction are collected
and measured following the same trapping and separation steps as in the 1st extraction. Note
that the procedure of the 2nd extraction can be repeated any number of times (e.g. 3rd extraction).

The amount of gases that we obtain from the 1st extraction comprises the atmospheric amount,
a possible contribution by in situ production, and a potential time-dependent production/release
in the melt water (*in extractu*). The 2nd extraction, however, targets only the *in extractu* fraction.
The system blank for the 2nd extraction was quantified using very clean Antarctic ice (Talos
Dome, EDC) and is < 1% of the amount of extracted species in the ice extraction and lower
than the measurement uncertainty.
Due to the small amount of $CH_4$ analyzed in this 2nd extraction (about a factor of 20 to 50 less
than for an ice core sample) the precision for the $\delta^{13}C$ analysis is much lower than for the 1st
(ice core) extraction and we estimate the precision of $\delta^{13}C\text{-}CH_4$ to 2 ‰ and for $[CH_4]$ to be 2
ppb or 10 %. For $C_2H_6$ and $C_3H_8$, the precision is comparable to the 1st extraction. The blank
values analyzed were 2 ppb, 0.3 ppb and 0.3 ppb for $CH_4$, $C_2H_6$ and $C_3H_8$, respectively,
assuming an ice core sample air volume of 14 mL at standard temperature and pressure, which
is the typical ice sample size of 150 g with a total air content of 0.09 mL/g.



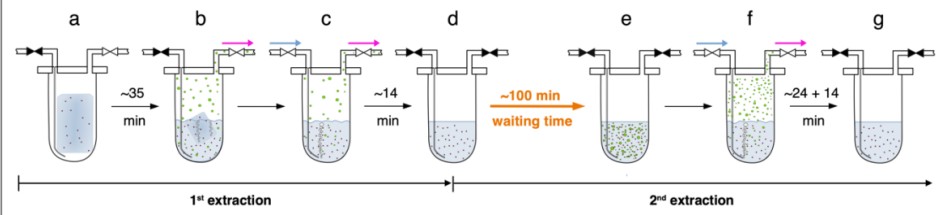


Figure 3: **Sequential steps (a-g) happening in the ice core sample vessel during the 1st and the 2nd extraction**
**in the $\delta^{13}C\text{-}CH_4$ extraction line**. Scheme illustrates the subsequent steps as described in detail in the text.
Brownish spots indicate dust particles in the ice/ melt water. Green circles indicate gas species (methane, ethane,
and propane) in the melt water or in the headspace of the vessel. Closed valves are indicated in black, open vales
in white. Blue arrows indicate the He flow through the inlet capillary into the sample vessel, pink arrows indicate
the flow direction from the sample vessel towards the AirTrap.






**2.3 $\delta$D-CH$_4$ Analysis of Ice Core Samples**

All $\delta$D-CH$_4$ data presented here were measured at the University of Bern using the discrete wet
extraction technique as described in detail in Bock et al. (2010a, 2014). This $\delta$D-CH$_4$ device
allows to measure the concentration of methane and its deuterium isotopic signature ($\delta$D-CH$_4$).

Briefly, ice core samples are melted after evacuation of the headspace using a warm water bath
at 40°C for 25-30 min to release the enclosed air into the sample vessel headspace. Once all the
ice is melted, the warm water bath is replaced by an ice-water bath to keep the melt water
temperature and water vapor pressure low. Note, in contrast to the $\delta^{13}$C-CH$_4$ method, the inlet
and outlet valves are closed during the melting process. The released air leads to an increased
pressure in the sample vessel headspace enhancing the solubility of gases in water.
Consecutively, the inlet and outlet valves are opened and He is purged for ~40 min with a flow
of 360 mL/min to transfer the accumulated air in the headspace and bubble He through the melt
water to strip dissolved gases. As for the $\delta^{13}$C-CH$_4$ method, the air is collected on an activated
carbon trap followed by further purification steps including GC separation. Note that compared
to the $\delta^{13}$C-CH$_4$ device, we performed only one extraction with the $\delta$D-CH$_4$ device.
For both methods, we assume that the time for an *in extractu* production during the ice
extraction procedure starts with the first presence of melt water until He purging is stopped.
Note that this time is considerably longer for the $\delta$D-CH$_4$ analysis (~90 min) compared to the
time of the 1st extraction in the $\delta^{13}$C-CH$_4$ analysis (~35 min).

Using this method we can measure [CH$_4$] and $\delta$D-CH$_4$ with a precision of about 15 ppb and 3
‰ (based on standard ice sample measurements), where isotopic data are expressed using the
$\delta$ notation on the international Standard Mean Ocean Water (SMOW) scale.



**3. Characterization of excess alkanes in ice cores**
**3.1 Methane, ethane, propane concentrations**

As described in detail in Sect. 2.2 a full ice sample measurement includes the regular ice sample
extraction (1st extraction) and, after the waiting time of ~100 min, a 2nd gas extraction in the
melt water. Gas from the 1st extraction is comprised of atmospheric air, a possible contribution
from in situ production, a potential time-dependent contribution by an *in extractu* process, and




any contribution from the device itself (blank). For the gas species discussed here (methane,
ethane, propane), these individual fractions are very different in magnitude. For polar ice core
samples, the atmospheric air is the major fraction of methane even in dusty, glacial ice from
Greenland prone to $CH_{4(xs)}$ production (see below). The opposite is true for ethane and propane,
which are dominated by the *in extractu* component in dust-rich Greenland ice. To establish a
better knowledge of alkanes in Greenland ice, we evaluated the measured concentrations of
methane, ethane, and propane, their ratios to each other and the relation to the content of mineral
dust in the ice with respect to the 1$^{st}$ and the 2$^{nd}$ extraction.
Note that different units to indicate concentrations of the trace gases of interest are used
throughout this study. By using mixing ratios in units of [ppb], as typically used for atmospheric
concentrations, the concentration of trace gases is related to the amount of air included in the
ice. Ice core samples with a low air content cause higher mixing ratio values for any additional
molecules produced in situ or *in extractu* compared to ice core samples with a high air content
and the interpretation might be biased. Alternatively, for any additional molecules produced in
situ or *in extractu*, [mol absolute per sample] denotes the absolute amount of trace gases and is
independent of the ice core air content. In the following, both units are used and great care has
to be taken to avoid misinterpretation of the results with respect to the different units.


**3.1.1 Excess alkanes in the 1$^{st}$ extraction**

Figure 4 and 5 show results from the 1$^{st}$ extraction of our NGRIP and GRIP ice core samples.
For dust-rich samples, ethane ranges between 2 ppb and 12 ppb, and propane concentrations
between 1 ppb and 5 ppb. In contrast, low-dust samples from both GRIP and NGRIP have much
lower concentration (ca. 0.5 ppb for ethane, and 0.3 ppb for propane) consistent with estimates
of past atmospheric ethane and propane concentrations from the 15$^{th}$ to 19$^{th}$ century of the
common era being about 0.4 ppb over Greenland (Nicewonger at al., 2016) and lower for
propane (Helmig et al., 2013). Emissions of ethane and propane were likely not drastically
larger during the glacial (Bock et al., 2017; Nicewonger et al., 2016; Dyonisius et al., 2020)
thus, 0.5 ppb appears to be an upper limit of past atmospheric concentrations of ethane and
propane. This estimate of past atmospheric ethane concentrations is an order of magnitude
smaller than the values we obtained from our dust-rich ice core samples from the 1$^{st}$ extraction,
pointing to an additional source of these alkanes for dust-rich samples.





As illustrated in Fig. 5 (left panel), the ethane and propane concentrations are highly correlated,
pointing to a common production of excess ethane and excess propane. The weighted mean
ratio (weighted according to the number of samples measured per bag) and its weighted
standard deviation (calculated by Gaussian error propagation of the weighted mean) is (2.25 ±
0.09) ppb ethane/ ppb propane. In Fig. 5, where the individual bags studied are color-coded, we
can clearly see that the ratio is essentially the same between the individual bags and that the
correlation is also very high within each bag (although we have to consider for the significance
of this correlation that the number of samples per bag is very low). This indicates that for
NGRIP ice ethane and propane are found in a fixed ratio. Accordingly, excess ethane and
propane production can be well represented by the weighted mean ratio and ethane and propane
are produced in a ratio of approximately 2:1. Very similar results were also observed in NGRIP
samples measured in 2011 and in GRIP samples revealing an ethane to propane ratio of 2.14 ±
0.03 ($r^2 = 0.99$) and 2.00 ± 0.13 ($r^2 = 0.99$), respectively (see Fig. 5, left panel).
Note that for a coherent presentation throughout the paper and a better comparison, ethane is
always plotted on the y-axis while we partly discuss ratios the other way round.

Methane concentrations range from 407 ppb to 476 ppb and are predominantly of atmospheric
origin. The amount of $CH_{4(xs)}$ is the difference between the measured methane concentration
and the atmospheric background concentration. To quantify $CH_{4(xs)}$ we use the fact that due to
the low-pass filtering of the bubble enclosure process all samples within one bag should have
the same atmospheric $CH_4$ concentration. This also ensures that any physical processes that
potentially influence the atmospheric alkanes in our samples (gravitational enrichment,
thermodiffusion, disequilibrium effects on $CH_4$ isotopes) are the same for all samples within
one bag. The only difference between these samples is, thus, the degree of $CH_{4(xs)}$ production
which is calculated from the linear fit between the measured $CH_4$ concentration and the
concentration of another species (e.g. ethane, propane, mineral dust, or $Ca^{2+}$), which serves as
a proxy for $CH_{4(xs)}$ production. The most precise relationship was found for $[C_2H_6]$ and
quantifying $CH_{4(xs)}$ was done by extrapolating the linear regression between ethane and methane
to an ethane concentration of 0.39 ppb, the assumed atmospheric $[C_2H_6]$. This leads to an
estimate of the true atmospheric $[CH_4]$ value within the respective bag, a value that can then be
subtracted from the measured $CH_4$ concentration to obtain the $CH_{4(xs)}$ in each sample. The
uncertainty of the calculated $CH_{4(xs)}$ is typically 8 ppb. Using the relation of ethane to methane
this approach translates into $CH_{4(xs)}$ in the range of 14 ppb to 91 ppb for these five NGRIP bags
with a mean excess of 39 ppb. Note, this mean value is not representative for this time interval



as values are biased towards higher values as we intentionally selected samples with high $Ca^{2+}$
content for our study. Equivalent calculations can be made using propane, dust, or $Ca^{2+}$ as proxy
for $CH_{4(xs)}$ production, however, the relationship between dust parameters and $CH_{4(xs)}$ is more
variable and does not lead to equally precise values for $CH_{4(xs)}$. Nevertheless, the obtained mean
$CH_{4(xs)}$ using the relation of mineral dust or $Ca^{2+}$ to methane is similar to the one obtained by
ethane.
We find that there is a constant production ratio between the measured excess alkanes.
Production ratios are the average of single-bag ratios weighted by the numbers of samples
measured per bag. Alkane concentrations were highly correlated within single-bags. The
weighted mean ratio and its weighted standard deviation was calculated to be $(6.42 \pm 1.57)$ ppb
methane / ppb ethane and $(14.3 \pm 3.7)$ ppb methane/ ppb propane for the samples of the five
main NGRIP bags, and $(2.25 \pm 0.09)$ ppb ethane/ ppb propane (also including NGRIP2011 and
GRIP here). We therefore characterize our measured NGRIP samples with an overall
methane/ethane/propane ratio of approximately 14:2:1. This constant relationship between
different alkanes suggests that excess alkanes are produced in a fixed ratio by a common
production process.
Another important observation is the close relation between excess alkanes and the content of
mineral dust within the ice core samples. Using measurements on GISP2 and NEEM ice core
samples, Lee et al. (2020) reported for the first time the close relation of $CH_{4(xs)}$ to chemical
impurities with the highest correlation with $Ca^{2+}$. This is supported by our measurements on
NGRIP and GRIP samples revealing on overall increase of $CH_{4(xs)}$ as well as ethane and
propane with increasing $Ca^{2+}$ (Fig. 5, right panel). Although the connection between ethane and
$Ca^{2+}$ is more variable than for ethane and propane between the different bags, the slopes of the
linear regressions in Fig. 5 (right panel) are still the same within the 2 σ uncertainty and the
weighted mean ratio of all NGRIP samples amounts to $(0.0089 \pm 0.0024)$ ppb ethane/ (ng/g)
$Ca^{2+}$.
However, this weighted mean value is likely biased low due to the relatively low ethane/ $Ca^{2+}$
slope of bag 3515. Due to a data gap at 1932.7 m in the $Ca^{2+}$ record, the corresponding $Ca^{2+}$
concentration for two of the samples of this bag is subject to a large interpolation error and
overestimated $Ca^{2+}$ (see Fig. A3). Note also the mismatch in the peak shape of the $Ca^{2+}$ and that
of the dust mass suggesting an anomalous aerosol chemistry for this peak.
These results agree with results from GRIP and older NGRIP (2011) samples, revealing an
ethane/ $Ca^{2+}$ ratio of $0.0105 \pm 0.0029$ ($r^2 = 0.76$) and $0.0090 \pm 0.0006$ ($r^2 = 0.91$), respectively.



Based on the fixed ratio of excess $CH_4$ and ethane described above this translates into a
weighted mean excess $CH_4/Ca^{2+}$ ratio of $(0.0529 \pm 0.0111)$ ppb methane per (ng/g) $Ca^{2+}$. Note
that due to the larger variability in the excess $CH_4$/ethane variation and the substantial
variability in the ethane/$Ca^{2+}$ relationship the relative uncertainty of this excess $CH_4/Ca^{2+}$
relationship is relatively large and dust and $Ca^{2+}$ are less suitable proxies to estimate $CH_{4(xs)}$
compared to ethane or propane.

Taken these findings together, we see a constant relationship between excess methane, ethane,
and propane, but also a close relation to the content of mineral dust within the ice core sample,
which, however, is not as tight as for the alkanes and suggests that dust parameters are only an
indirect proxy of the alkane excess.

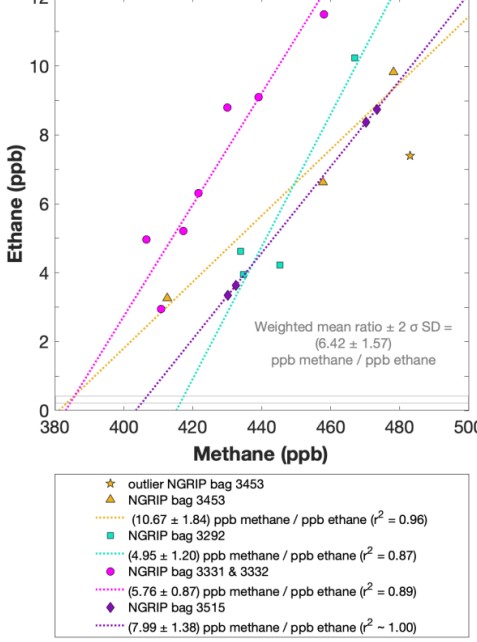



Figure 4: **NGRIP results of methane and ethane from the 1ˢᵗ extraction**. Concentrations of methane (ppb) and
ethane (ppb) and their ratios to each other for NGRIP samples measured in the 1ˢᵗ extraction of the $\delta^{13}C$-$CH_4$
device. Different colors and symbols indicate the different NGRIP bags used for our analysis. Note that there is
an outlier for $CH_4$ in bag 3453 as indicated in a yellow asterix, which is not included in the ratio of bag 3453. The
grey hatched area indicates past atmospheric ethane concentrations of maximum 0.39 ppb as estimated by
Nicewonger et al. (2016).







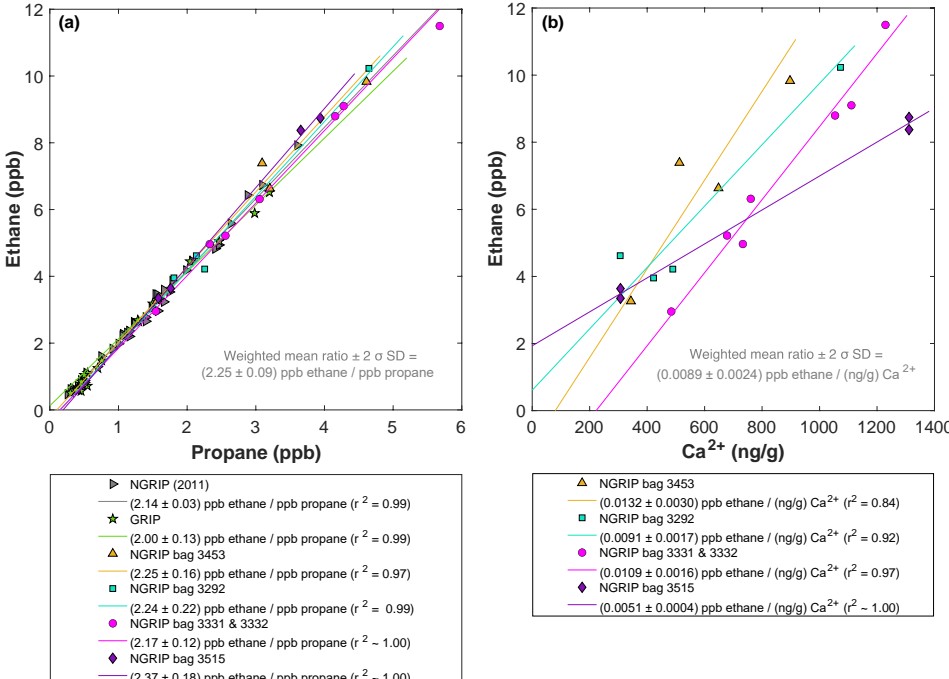


Figure 5: **NGRIP and GRIP results of ethane and propane from the 1$^{st}$ extraction. (a)** Concentrations of ethane and propane and their ratios to each other for NGRIP and GRIP samples measured in the 1$^{st}$ extraction of the $\delta^{13}$C-CH$_4$ device. Colors and symbols indicate the different NGRIP bags or cores used. **(b)** Bag-specific production ratios of ethane in relation to the Ca$^{2+}$ concentration for NGRIP samples. Note that for bag 3515 there is a data gap and an anomaly of the Ca$^{2+}$ to dust mass ratio for the replicate sample at 1932.7 m and the Ca$^{2+}$ concentration for these two data points is likely overestimated by a few 100 ng/g (see Fig. A3). Thus, the ethane/ Ca$^{2+}$ slope for this bag is likely biased toward too high values.

486

487

### 3.1.2 Excess alkanes in the 2$^{nd}$ extraction

489

With the 2$^{nd}$ extraction in the $\delta^{13}$C-CH$_4$ analyses we can evaluate the temporal dynamics of excess alkane production, assuming that gas extraction during the 1$^{st}$ extraction was quantitative and all alkanes extracted in the 2$^{nd}$ extraction were produced in the time after the 1$^{st}$ extraction was completed.

For our Greenland samples we measured a range of about 0.2 to 2.4 pmol for ethane and a range of 0.1 to 1.2 pmol for propane in the 2$^{nd}$ extraction. These values in pmol are equivalent to 0.2 to 48 ppb ethane and 0.2 to 2 ppb propane assuming that the amount of excess alkanes was added to 14 mL of ice core air (which is the typical ice sample size of 150 g with a total air content of 0.09 mL/g) (Fig. 6, right panel). The measured amount of methane ranges between





3 pmol and 20 pmol (Fig. 6, left panel). The ratio of the measured amount for the individual
species between the 1$^{st}$ and the 2$^{nd}$ extraction amounts to $3.6 \pm 0.85$ ($r^2 = 0.78$) for ethane (Fig.
7, right panel), $3.3 \pm 0.33$ ($r^2 = 0.78$) for propane (combined data of NGRIP and GRIP) and 3.8
$\pm 1.62$ ($r^2 = 0.33$) for methane (only NGRIP data), where the uncertainty for $CH_4$ is again much
larger.
Thus, we can conclude that the amount of alkanes produced during the waiting time after the
1$^{st}$ extraction until the 2$^{nd}$ extraction was finished, was approximately 30% of the amount
produced during the 1$^{st}$ extraction.
We can therefore safely conclude that excess alkanes are also produced/ released during the 2$^{nd}$
extraction. Results from the 2$^{nd}$ extraction also demonstrate that this process is slow and not
completed during the time of the 1$^{st}$ extraction. We can thereby confirm the results of Lee et al.
(2020) and here we are able to show for the first time that this process leads also to production
of excess ethane and propane.

For a better estimate of the temporal reaction kinetics of the underlying process, we can relate
the measured amount of the individual species to the time available for a potential reaction in
the melt water during each extraction. For the five GRIP samples that were measured with a 2$^{nd}$
and 3$^{rd}$ extraction (see Sec. 2.1 and 2.2 for details) we take the cumulative production amount
(where the first data point is the produced amount in the 1$^{st}$ extraction, the second data point is
the sum of the 1$^{st}$ and 2$^{nd}$ extraction, and the third data point is the sum of the 1$^{st}$, 2$^{nd}$, and 3$^{rd}$
extraction). Exemplarily shown for ethane (Fig. B1, Appendix B) we can see the assumed first-
order reaction kinetics with an exponential accumulation of ethane over time (accompanied by
an exponential decay of organic precursor substances) providing a good model for our
measurements. With that, we can estimate the half-life time ($\tau$) of the production to be
approximately 30 min. Compared to continuous flow techniques, where the reaction time before
the air is separated from the liquid water stream, is only 1-2 min, only 5-10 % of the *in extractu*
production found in our 1$^{st}$ extraction can be expected.

The goodness of fit of the ratios of the measured concentrations between the 1$^{st}$ and the 2$^{nd}$
extraction is $r^2 = 0.78$ for both ethane and propane, indicating that the production/release in the
1$^{st}$ extraction in relation to the 2$^{nd}$ extraction is well correlated for both species (Fig. 7b). Thus,
samples that produced higher excess alkanes during the 1$^{st}$ extraction also produced more
excess alkanes in the 2$^{nd}$ extraction suggesting that the production is dependent on the amount
of some reactant present in the samples from which excess alkanes are produced. Again, for





$CH_4$ this relationship is more variable which is likely related to the higher uncertainty in
measuring $CH_4$ for the 2nd extraction.

The ratio of ethane to propane of all measured Greenland samples in the 2nd extraction is 2.00
$\pm$ 0.07 ( $r^2$ = 0.99). The ratio of methane to ethane is 8.34 $\pm$ 1.07 ($r^2$ = 0.93). Accordingly, the
overall relationship between methane, ethane, and propane in the 2nd extraction can be
characterized by a ratio of approximately 16:2:1. Comparing the ratios of ethane/ propane and
methane/ ethane between the 1st and the 2nd extraction, there is no significant difference within
the 2 $\sigma$ uncertainties from 2.25 $\pm$ 0.09 to 2.00 $\pm$ 0.07, and from 6.42 $\pm$ 1.57 to 8.34 $\pm$ 1.07. We
can conclude that within the error limits, the ratios stayed the same suggesting that the same *in*
*extractu* process is at play during both extractions.

In the 2nd extraction, we can again observe the relation between excess alkanes and the amount
of mineral dust. Figure 7 (left panel) shows the correlation of ethane (fmol/g melt water) to
$Ca^{2+}$ (ng/g) in all measured NGRIP and GRIP samples in the 2nd extraction revealing a
production of (0.0085 $\pm$ 0.0011) fmol/(g melt water) ethane per (ng/g) $Ca^{2+}$ with $r^2$ = 0.70. For
methane, we observe a production ratio of (0.0556 $\pm$ 0.01513) fmol/(g melt water) methane per
(ng/g) $Ca^{2+}$ with a correlation of $r^2$ = 0.47 (data not shown).

Overall, excess alkane concentrations are increasing with increasing $Ca^{2+}$ concentrations, in
both the 1st and the 2nd extraction. The alkane production/release, however, decreased in the 2nd
extraction, suggesting the progressive exhaustion over time of some reactant necessary for the
*in extractu* process. We propose that this reactant co-varies with $Ca^{2+}$ and particulate dust and
that $Ca^{2+}$ concentrations are only a proxy for higher *in extractu* production.




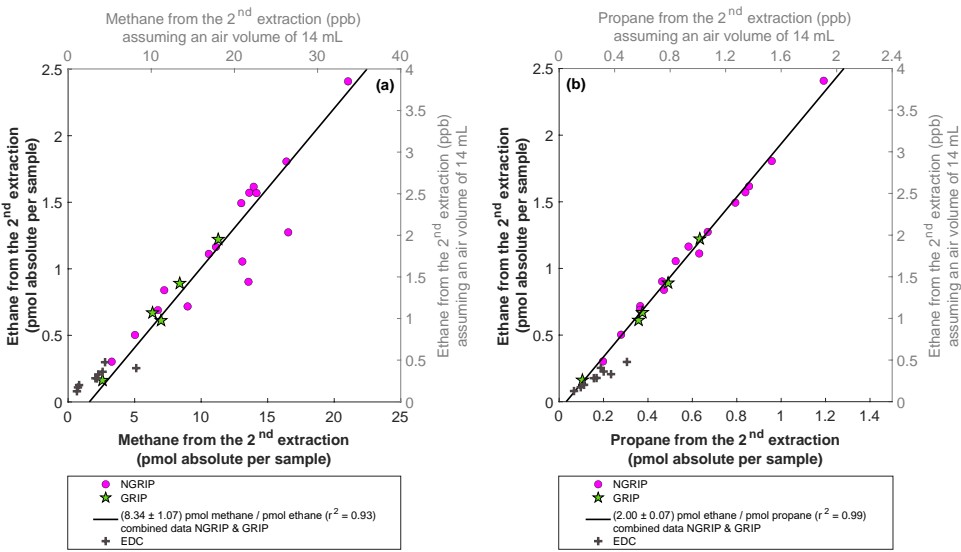


Figure 6: **NGRIP and GRIP results of excess methane, ethane, and propane from the 2nd extraction. (a)**
Concentrations of methane and ethane and their ratios to each other. **(b)** Concentrations of propane and ethane and
their ratios to each other. Units are given as pmol absolute per sample on the primary axis in black and in ppb
assuming an air volume of 14 mL of the ice core sample on the secondary axis in grey. Grey crosses indicate the
blank level of the system estimated from EDC ice core sample measurements.


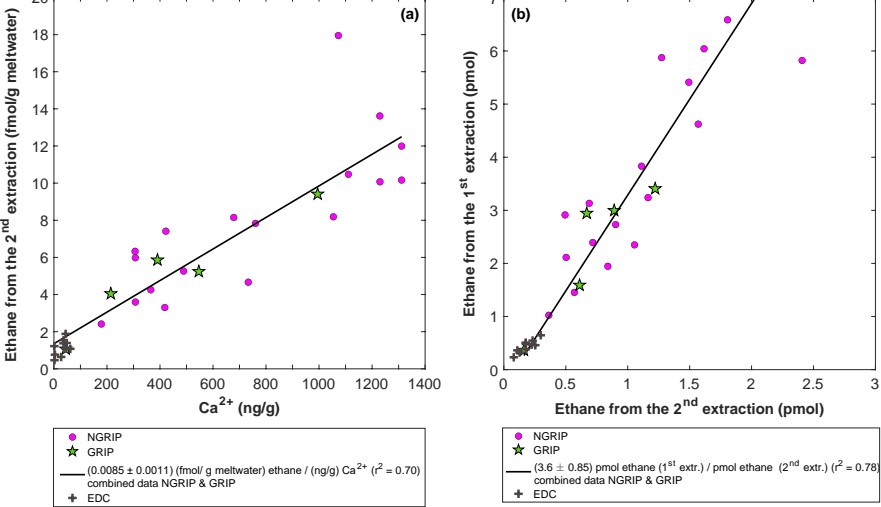

Figure 7: **NGRIP and GRIP results of ethane from the 2nd extraction in relation to the Ca²⁺ concentration**
**and to the 1st extraction. (a)** Produced amount of ethane in the meltwater in relation to the Ca²⁺ concentration
within the ice core samples. Grey crosses indicate the blank level of the system estimated from EDC ice core
sample measurements. **(b)** Relation of the amount of ethane measured in the 1st and 2nd extraction.



**3.2 Isotopic composition of excess methane**

The evaluation of the carbon and deuterium isotopic signature of excess methane ($\delta^{13}$C-CH$_{4(xs)}$ and $\delta$D-CH$_{4(xs)}$) is based on the Keeling-plot approach (Keeling, 1958, 1961; Köhler et al., 2006). Here, we want to characterize the isotopic signature of excess methane and explore how we can use this parameter to better identify its source or production pathway.

**3.2.1 $\delta^{13}$C-CH$_4$ isotopic signature of excess methane**

Figure 8 (left panel) shows the $\delta^{13}$C-CH$_4$ results of the 1$^{st}$ extraction. The carbon isotopic signature of excess CH$_4$ from the 1$^{st}$ extraction of the ice core sample measurements within one NGRIP bag are obtained from the y-intercept of the Keeling-plot, representing the excess $\delta^{13}$C-CH$_4$ value for this bag. All bags show agreement in $\delta^{13}$C-CH$_4$ signature (y-intercepts) within the 2 $\sigma$ uncertainties. The weighted mean isotopic signature is (-46.4 $\pm$ 2.4) ‰, with weights assigned by the number of samples that constrained each individual Keeling plot regression line.

Figure 8 (right panel) shows the isotopic results in relation to the amount of CH$_4$ produced during the 2$^{nd}$ extraction. No atmospheric CH$_4$ is present during the 2$^{nd}$ extraction and the individual isotopic values in Fig. 8 (right panel) are the directly measured values of excess CH$_4$ without applying the Keeling-plot approach. For a better comparison, the produced CH$_4$ is shown both in pmol (lower axis in Fig. 8, right panel) and in a mixing ratio CH$_4$ scale (ppb), where we assume that the excess CH$_4$ produced during the 2$^{nd}$ extraction is diluted into an air volume of 14 mL at standard temperature and pressure, which is a typical value for the amount of air extracted from our samples in the 1$^{st}$ extraction.

The Keeling y-intercept values of the 1$^{st}$ extraction are added in the right panel of Fig. 8.

The $\delta^{13}$C-CH$_4$ values of the 2$^{nd}$ extraction range between -34 ‰ and -48 ‰ with the mean being (-41.2 $\pm$ 2.2) ‰. This value is isotopically heavier compared to the weighted mean of (-46.4 $\pm$ 2.4) ‰ inferred from the Keeling analysis, however, is within the 2 $\sigma$ error limits. We note that the measured peak areas for the 2$^{nd}$ extractions are very small and lie outside of the typical range of our gas chromatography mass spectrometry analysis for $\delta^{13}$C-CH$_4$ and we cannot exclude some bias in these results. However, we mimicked these small peak areas with injections of small amounts of standard air and observed no significant bias in the measured





$\delta^{13}C$-$CH_4$ values given that the precision of such small peaks is around 2 ‰. Another caveat is
the considerable blank contribution for $CH_4$ that we observe for the 2$^{nd}$ extraction. Since
Antarctic ice cores do not show a sizable *in extractu* production (Fig. 7, grey crosses for EDC)
we measured EDC samples with the same protocol as for our Greenland samples to provide an
upper boundary of this blank. As can be seen in Fig. 8 (right panel) the amount of $CH_4$ measured
for these EDC samples (grey crosses) is on average about 2 pmol (equivalent to about 2 ppb).
For comparison, our ice samples from Greenland show a range of about 5 to 20 pmol, thus we
have a considerable blank contribution. However, the $\delta^{13}C$-$CH_4$ blank signature obtained from
these EDC samples is comparable to or only a few ‰ heavier than the $\delta^{13}C$-$CH_4$ signature of
the excess $CH_4$ from this 2$^{nd}$ extraction for the Greenland samples. Considering these analytical
limitations of our 2$^{nd}$ extraction for $\delta^{13}C$-$CH_4$, these findings suggest that excess $CH_4$ produced
during the 1$^{st}$ and 2$^{nd}$ extraction has a similar $\delta^{13}C$-$CH_4$ isotopic signature and is likely
produced/released by the same process in both extractions.

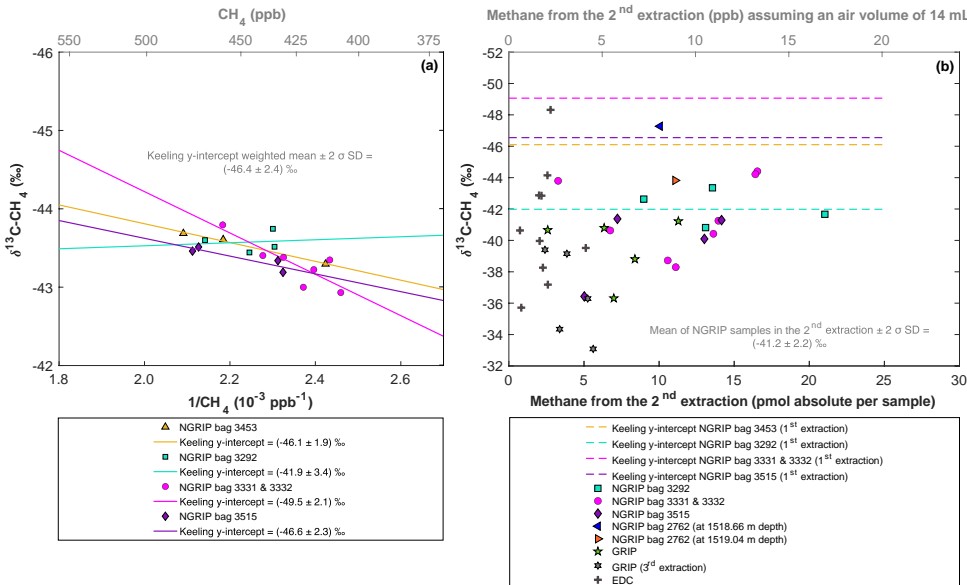



Figure 8: **NGRIP (and GRIP) $\delta^{13}C$-$CH_4$ results of the 1$^{st}$ and 2$^{nd}$ extraction measured with the $\delta^{13}C$-$CH_4$**
**device. (a)** Keeling-plot of $\delta^{13}C$-$CH_4$ for NGRIP samples from the five main bags (3292, 3331 & 3332, 3453,
3515) measured in the 1$^{st}$ extraction. Colors and symbols indicate individual measurements of the respective bags.
Colored lines indicate the corresponding Keeling regression line of each individual bag. **(b)** $\delta^{13}C$-$CH_4$ (‰) values
in relation to the amount of methane measured in the 2$^{nd}$ extraction. Units for $CH_4$ are given as pmol absolute per
sample on the primary axis in black, and in ppb assuming an air volume of 14mL of an ice core sample on the
secondary axis in grey. Colors and symbols indicate individual measurements of the respective bags. Color-coded
lines indicate the corresponding Keeling y-intercept of each individual bag as measured in the 1$^{st}$ extraction. Grey
crosses indicate the blank level estimated from EDC ice core measurements.



**3.2.2 $\delta$D-CH$_4$ isotopic signature of excess methane**

Figure 9 shows the isotopic results of the $\delta$D-CH$_4$ analyses. Due to the larger sample size
required for the $\delta$D-CH$_4$ analyses and the sample availability restrictions only two bags were
studied for $\delta$D-CH$_4$. The individual isotopic results obtained from the ice core sample
measurements within one NGRIP bag are again combined into one Keeling y-intercept,
representing the $\delta$D-CH$_4$ value for this bag. NGRIP bag 3460 (orange) reveals a Keeling y-
intercept $\delta$D-CH$_4$ value of (-308 ± 51) ‰. The two NGRIP bags 3266 and 3267 (purple) are
neighbouring bags and were therefore combined into one Keeling y-intercept revealing a $\delta$D-
CH$_4$ value of (341 ± 62) ‰. The difference between the two Keeling y-intercepts of the
individual bags is within the error limits and thus do not show significant differences.
Accordingly, we combine the two values to a weighted mean and weighted uncertainty of
(-326 ± 57) ‰.
Our results are consistent with the findings of Lee et al. (2020), who used the NGRIP $\delta$D-CH$_4$
record of Bock et al. (2010b) and the NGRIP [CH$_4$] record of Baumgartner at al. (2014) to
estimate the $\delta$D-CH$_{4(xs)}$ signature in these samples. Assuming a two-component mixture of
atmospheric methane and excess methane in their model led to a best estimate of (-293 ± 31)
‰ for $\delta$D-CH$_{4(xs)}$ which is within the error limits of our Keeling-plot results.

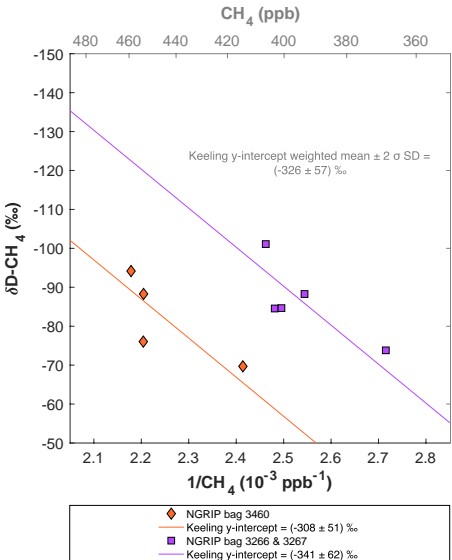


Figure 9: **GRIP $\delta$D-CH$_4$ results.** Keeling-plot of $\delta$D-CH$_4$ of NGRIP samples measured with the $\delta$D-CH$_4$ device.
Colors and symbols indicate individual measurements of the respective bags and colored lines indicate the
corresponding regression of each individual bag.



**4. Testing the hypotheses explaining excess alkanes**

In Sect. 3 several pieces of evidence for the production/release of excess alkanes in Greenland ice core samples were collected:

- We can confirm the observations of Lee et al. (2020) on excess methane in different Greenland ice cores and its covariance with the amount of mineral dust in the ice. Despite the different extraction techniques applied (multiple melt-refreeze method in Lee et al. (2020) versus two subsequent wet extractions in our study), we can further corroborate that the temporal dynamics of the production/ release is on the order of hours and production/ release occurs when liquid water is present during extraction.
- We document for the first time a co-production/ release of excess methane, ethane, and propane, with the observed values for ethane and propane exceeding by far their estimated past atmospheric background concentrations.
- Excess alkanes (methane, ethane, propane) are produced/ released in a fixed molar ratio of approximately 14:2:1, indicating a common origin.
- We further characterize the isotopic composition of excess $CH_4$ of $\delta^{13}C$-$CH_{4(xs)}$ and $\delta D$-$CH_{4(xs)}$ to be (-46.4 ± 2.4) ‰ and (-326 ± 57) ‰ in NGRIP ice core samples, respectively. Within error limits, our $\delta D$-$CH_{4(xs)}$ results are consistent with the calculated best estimate of (-293 ± 31) ‰ by Lee et al. (2020).

In the introduction we presented the hypotheses proposed by Lee et al. (2020) explaining their observations on $CH_{4(xs)}$. Here we resume the discussion of the original hypotheses and refine them in light of our new data from NGRIP and GRIP ice sample measurements. An overview of the different possible sources explaining excess alkanes is illustrated in Fig. 10. We believe that the origin of the observed excess alkanes falls in one of the three categories:

1.) Excess alkanes could be adsorbed on mineral dust particles prior to their deposition on the Greenland ice sheet and released in the laboratory during the prolonged melting process. The adsorption step could happen in the mineral dust source region (East Asian deserts) thereby adsorbing the alkanes from natural gas seeps within the sediment (process marked as A1, see Fig. 10). Alternatively, there is adsorption of atmospheric alkanes on dust particles either at the soil surface in the dust source region or during atmospheric transport to the Greenland ice sheet after deflation (A2). The desorption of the adsorbed alkanes happens during the melting process for both cases.



2.) Excess alkanes could be produced microbially. The production happens either in the ice
itself (in situ) and the alkanes are then subsequently released during the melting phase in the
laboratory (M1). Alternatively, the microbial production happens in the melt water during the
melting process (*in extractu)* (M2). A microbial in situ production in the ice without an
adsorption-desorption process was already ruled out by Lee et al. (2020) since it is not
compatible with evidence from the CFA CH$_4$ concentration records. Excess CH$_4$ is not observed
in CFA records implying that the extraction/production of excess alkanes is slow relative to the
short extraction time of CFA. This was used as evidence for desorption of alkanes from mineral
dust particles in the ice which would be released slowly at the presence of liquid water and
effect techniques using longer extractions.
3.) Excess alkanes are produced abiotically, e.g. by the decomposition of labile organic
compounds. This chemical reaction can happen either in the ice itself (in situ), which are then
adsorbed on dust particles and subsequently released during the melting process (C1), or in the
melt water during extraction (*in extractu*) (C2). An abiotic in situ production in the ice without
an adsorption-desorption process can also be ruled out with the CFA evidence.

We now discuss these mechanisms in detail and evaluate the viability of the different
hypotheses in the light of our new experimental observations.







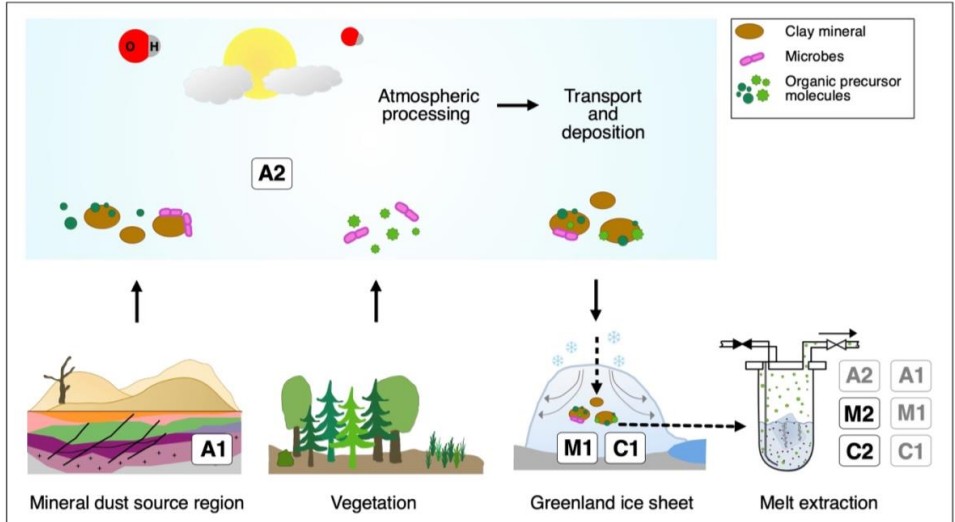


Figure 10: **Overview of the different possibilities explaining excess alkanes in dusty Greenland ice**. **A** depicts
an adsorption process of alkanes on mineral particles, either from natural gas seeps within the sediment (A1) or
from the atmosphere (A2) prior to their deposition on the Greenland ice sheet and in relation to a subsequent
desorption process during the melting process in the laboratory. **M** depicts a microbial production of excess
alkanes, either in the ice itself (in situ), and in relation to an adsorption process on dust particles after production
in the ice and a subsequent desorption process during the melting process (M1), or in the melt water (*in extractu*)
(M2). **C** depicts the abiotic/ chemical production of excess alkanes, either in the ice itself (in situ), and in relation
to an adsorption process on dust particles after production in the ice and a subsequent desorption during the melting
process (C1), or in the melt water (*in extractu*) (C2).

**(1) Adsorption/release of alkanes on mineral dust particles**
In the following section we discuss the mechanism to explain our observations which are based
on the adsorption of excess alkanes onto mineral dust particles. Depending on where the
adsorption takes place, the mineral particles might adsorb alkanes of different origin and
composition. One possibility is that the adsorption already takes place within the sediment or
soil of the dust source region, thus before deflation (A1). As proposed by Lee et al. (2020), the
major source region of mineral dust arriving in Greenland during the glacial (Taklamakan,
Tarim Basin) are also regions where natural gas seeps reach the surface (Etiope and Klusman,
2002; Etiope et al., 2008). Alternatively, the dust particles adsorb alkanes that are present in the
atmosphere and the adsorption can either happen at the soil surface in the dust source region or
en route to the Greenland ice sheet after deflation (A2). At first order, for the scenario A2 the
fingerprint (isotopic composition and ratio of alkanes) of the adsorbed alkanes depends on the



past atmospheric composition but could be modulated by selective fractionation processes
during adsorption.

To be a viable mechanism for our problem, it requires that the adsorbed alkanes stay strongly
bound at the mineral dust particles while desorption is insufficient both during the atmospheric
transport and during the several hundred years the dust particle spends in the porous firn (age
of the firn at bubble close off). During the melting procedure the adsorbed alkanes would then
be released from their mineral dust carrier, which in case of Greenland during glacial times is
predominately consisting of clay minerals from the Taklamakan (and partly also Gobi) desert
(Bory, et al., 2003, Svensson et al., 2000; Ruth et al., 2003; Rhodes et al., 2013). However,
other additional dust sources exist with their relative contribution varying with climate
conditions (Han et al., 2018; Lupker et al., 2010).

Evidence on the adsorptive capacity of alkanes on clay minerals and its strong retention was
accumulating from several experimental studies (Sugimoto et al., 2003; Cheng and Huang,
2004; Dan et al., 2004; Pires et al., 2008; Ross and Bustin, 2009; Ji et al., 2012; Liu et al., 2013;
Tian et al., 2017). While all clay minerals are expected to be $CH_4$ adsorbents (Sugimoto et al.,
2003), this was predominantly demonstrated for kaolinite, chlorite, illite, and montmorillonite
(Sugimoto et al., 2003; Cheng and Huang, 2004; Ross and Bustin, 2009; Ji et al., 2012; Liu et
al., 2013; Tian et al., 2017). Influencing parameters for an adsorption-desorption process are
mainly pressure, temperature, clay mineral type, micropore size, surface area, organic carbon
content, and water/ moisture content (Sugimoto et al., 2003; Cheng and Huang, 2004; Dan et
al., 2004; Pires et al., 2008; Ross and Bustin, 2009; Ji et al., 2012; Liu et al., 2013; Tian et al.,
2017). Most interestingly for us, studies by Sugimoto et al. (2003) and Dan et al. (2004) on the
adsorption of $CH_4$ in micropores on the surface of clay minerals in dried and fresh lake sediment
showed that dried sediment still retains $CH_4$ and that dried and degassed sediment re-adsorbs
ambient $CH_4$ at standard pressure of $CH_4$ and room temperature. The amount of $CH_4$ adsorbed
in their samples is strongly dependent on pressure and temperature while increasing
temperatures and decreasing pressure lead to a stronger desorption. The addition of water/
moisture leads to a rapid desorption of already adsorbed gases (Sugimoto et al., 2003; Dan et
al., 2004; Pires et al., 2008; Ji et al., 2012; Liu et al., 2013).

These results in principle support our hypothesis of an adsorption-desorption process for our
glacial NGRIP and GRIP ice core samples, where alkanes (from fossil seeps or atmosphere)





would be adsorbed on dust particles and desorbed during the measurement procedure when
liquid water is present. Independent of the origin of the alkanes (A1 or A2) the amount of
alkanes deposited onto the Greenland ice sheet by this process would be diminished if mineral
dust particles were already in contact with liquid water during the long-range transport which
may lead to a loss of previously adsorbed alkanes already in the atmosphere.

Regarding our experimental results, the high correlation between mineral dust ($Ca^{2+}$) and excess
alkanes observed in many Greenland ice cores would be generally in line with the theory of
adsorption on mineral dust. In our data we see that the amount of released excess alkanes per
$Ca^{2+}$ is variable (especially in the 2nd extraction), which can be explained by a varying
adsorption capacity of the mineral dust particles or a close relation between the adsorption
capacity and the type of clay mineral (Sugimoto et al., 2003; Ji et al., 2012). However, to explain
the constant ratio of methane, ethane, and propane of 14:2:1 in our samples with an adsorption
mechanism, we need to discuss the potential origins of the adsorbed alkanes.

First, we find very high relative excess contributions of ethane and propane in our samples,
while we see a small excess contribution for methane compared to the atmospheric background.
If we assume a comparable adsorption for all three alkanes, this would imply a strong relative
enrichment of ethane and propane over methane in the concentration of these gases during
adsorption. This is not in line with the past atmospheric $CH_4/(C_2H_6+C_3H_8)$ ratio where past
atmospheric ethane concentrations by Nicewonger et al. (2016) are an order of magnitude
smaller (and propane concentrations even less) than the measured concentrations in our NGRIP
and GRIP ice core samples. If we assume instead that excess alkanes have a thermogenic origin,
we see that the ratio of methane, ethane, and propane for our samples of approximately 14:2:1,
translated into a $CH_4/(C_2H_6+C_3H_8)$ ratio of ~5, is most consistent with a thermogenic origin,
albeit more at the lower limit (see Fig. 11, left panel). However, we also have to question a
selective adsorption capacity of mineral dust particles. If ethane and propane are preferentially
adsorbed over methane, this would misrepresent the actual ratio between the three alkanes and
falsify our interpretation of the origin.
To further evaluate the adsorption theory in the light of our experimental evidence, we now
include the carbon and deuterium isotopic signature of $CH_{4(xs)}$ in our samples. In our NGRIP
data we document a relatively heavy (enriched) $\delta^{13}C$-$CH_{4(xs)}$ signature (in both extractions),
which is close to the atmospheric value at that time but a light $\delta D$-$CH_{4(xs)}$ signature, which is
close to the typical microbial signature (see Fig. 11 for an overview of isotopic signatures and





alkane ratios). In general, atmospheric values for $\delta^{13}$C-CH$_4$ and $\delta$D-CH$_4$ are heavier in
atmospheric CH$_4$ compared to the global CH$_4$ source mix due to the fractionation by
atmospheric sink processes. Typical atmospheric values for the respective gas age of our
measured Greenland samples derived from Southern Hemisphere ice core samples are in the
range between -42 ‰ to -45 ‰ for $\delta^{13}$C-CH$_4$ (Möller et al., 2013) and between -50 ‰ to -80
‰ for $\delta$D-CH$_4$ (Möller et al., 2013; Bock et al., 2017; Dyonisius et al., 2020). Due to the
geographic distribution of sources and sinks, the true Greenland values at the respective ages
are a little lower. For example, the measured interhemispheric difference in $\delta^{13}$C-CH$_4$ over the
Holocene (Beck et al., 2018), when CH$_{4(xs)}$ is not observed, is less than 1 ‰ and given a
reduction of Northern Hemisphere sources during the Glacial, -42 ‰ to -45 ‰ can be regarded
as an upper limit for our glacial NGRIP samples. Thus, the EDML values provide a
representative first-order estimate of atmospheric $\delta^{13}$C-CH$_4$ also for our NGRIP ice core
samples.

In comparison to the atmospheric source mix, microbially produced CH$_4$ is depleted in both
heavy isotopologues ($^{13}$CH$_4$ and CH$_3$D) compared to the atmospheric value. Typical values for
microbial CH$_4$ are in the range between -150 ‰ to -450 ‰ for $\delta$D-CH$_4$ (Whiticar, 1999) and
between -55 ‰ to -70 ‰ for $\delta^{13}$C-CH$_4$ (see also Fig. 11, right panel). Thermogenic emissions
range between -25 ‰ to -55 ‰ for $\delta^{13}$C-CH$_4$ (Etiope and Klusman, 2002;) and between -100
‰ to -275 ‰ for $\delta$D-CH$_4$ (Whiticar, 1999; Etiope et al., 2007). Accordingly, while we expect
that any adsorbed CH$_4$ of atmospheric origin would at first-order reflect the atmospheric $\delta^{13}$C
and $\delta$D signature of CH$_4$ and would not be able to strongly affect the isotopic composition of
the CH$_4$ extracted from our ice core samples, CH$_4$ adsorbed at the dust source (e.g. thermogenic
origin) would not be subject to the fractionation by atmospheric sinks and would have a strong
leverage on the isotopic composition of extracted CH$_4$. For $\delta$D, where the atmospheric sink
fractionation is very strong, any CH$_{4(xs)}$ can therefore lower the $\delta$D signature of the ice core
sample drastically.

Our NGRIP samples reveal a $\delta^{13}$C-CH$_{4(xs)}$ value (Keeling y-intercept weighted mean) of (-46.4
± 2.4) ‰ which is within the error consistent with contemporaneous atmospheric values or with
emissions from seeping reservoirs of natural gas. In contrast, our hydrogen isotopic
measurements on NGRIP samples reveal a very light $\delta$D-CH$_{4(xs)}$ value (Keeling y-intercept
weighted mean) of (-326 ± 57) ‰ similar to the estimates by Lee at al. (2020) and outside of
the field of a thermogenic origin (see Fig. 11). While both the low CH$_4$/(C$_2$H$_6$+C$_3$H$_8$) ratio and



the $\delta^{13}C$-$CH_{4(xs)}$ could be indicative of a thermogenic source (A1), the light $\delta D$-$CH_{4(xs)}$ signature
is far away from the atmospheric $\delta D$-$CH_4$ value and is also not in line with typical $\delta D$-$CH_4$
values of a thermogenic origin. Hence, our $\delta D$-$CH_{4(xs)}$ values render the adsorption scenarios
A1 and A2 unrealistic candidates to explain our observations.


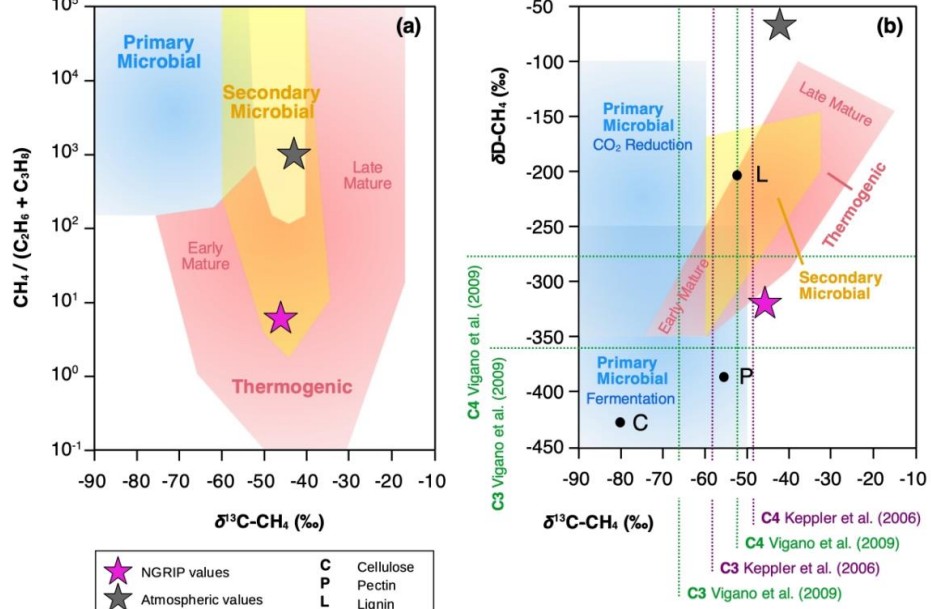


Figure 11: **Diagrams of genetic fields for natural gas adopted from Milkov and Etiope (2018). (a)** Genetic
diagram of $\delta^{13}C$-$CH_4$ versus $CH_4/(C_2H_6+C_3H_8)$. Typical atmospheric values are indicated with a grey star, NGRIP
values obtained from this study with a pink star. **(b)** Methane genetic diagram of $\delta^{13}C$-$CH_4$ versus $\delta D$-$CH_4$. Values
for cellulose (C), lignin (L) and pectin (P) from Vigano et al. (2009) and mean values for C3 and C4 plants,
respectively, from studies by Keppler et al. (2006) and Vigano et al. (2009) are added.



**(2) Microbial production**
The second process that we take into consideration regards the microbial production of excess
alkanes through methanogenic microbes. Here we must again differentiate between two
scenarios: a microbial production can either take place in the ice sheet itself (in situ) by
extremophile microbes. This process requires that in situ produced excess alkanes are then
adsorbed onto dust particles in the ice and subsequently desorbed during extraction when in




contact with liquid water (M1). Or the production takes place during the melt extraction when
methanogenic microbes can metabolize in liquid water (*in extractu*; M2). Lee et al. (2020)
already excluded a "simple" in situ production of excess $CH_4$ (microbial in situ production in
the ice without an adsorption-desorption process) and this option will therefore not be further
discussed here.

The viability of microbial in situ activity in the ice was substantially discussed in Lee et al.
(2020) and references therein. While there is evidence for high cell counts in association with
high concentrations of dust in Greenland ice cores, there is no direct evidence of active
methanogens capable of producing $CH_4$ in ice (Tung et al., 2005, 2006; Rohde et al., 2008;
Miteva et al., 2009). Calculations on the production of biogenic $CH_4$ in ice by Price and Sowers
(2004), Tung et al. (2005), and Rohde et al. (2008) lead to a best estimate of ~$5 \cdot 10^{-5}$ pmol $CH_4$/g
ice in 35 kyears. In comparison to our observations (for instance when taking the $CH_{4(xs)}$ mean
of ~32 ppb in the four samples measured in the NGRIP bag 3515 with a mean ice sample weight
of ~139 g) this translates into ~0.13 pmol $CH_4$/ g ice in 32 kyears, which is several magnitudes
higher. Moreover, we assume that in situ produced excess alkanes would increase with time
(depth) in relation to the amount of mineral dust within the ice until conditions no longer support
this process (i.e. nutrient limitation). This was tested by analyzing dust-rich GISP2 samples
ranging from 42-75 kyears, however, no time-dependent process was observed (Lee et al.,
2020). On the other hand, there are $CH_4$ anomalies in Greenland ice cores that might be caused
by microbial activity. Rhodes et al. (2013) report $CH_4$ spikes in the NEEM S1 core that are not
associated with melt events but are characterized by anomalously high concentration of $NH_4^+$
and other biomass burning-derived nutrients. Since these $CH_4$ spikes have been observed both
with the classic wet extraction and with the CFA technique that allows minimal reaction time
in liquid water during the melt phase, these $CH_4$ anomalies were likely produced already in the
ice, thus qualify as in situ. These narrow $CH_4$ spikes occur in Holocene ice with typically low
dust and $Ca^{2+}$ content, thus having a different impurity composition compared to our high-dust
samples where we observe *in extractu* alkanes. Similar $CH_4$ spikes without an association to
melt layers were reported in the GISP2 ice core by Mitchell et al. (2013).
Moreover, ice samples from different Greenland ice cores that are affected by melt events show
$CH_4$ anomalies as well (Rhodes et al., 2016; NEEM Community Members, 2013). Further
analyses are needed from these localized $CH_4$ spikes, e.g. if they show a co-production of ethane
and propane as well and if their origin is really in situ or *in extractu* as well but with a much



quicker reaction time that allows the reaction to be completed within a few minutes (rather than
hours in case of our *in extractu* phenomenon for dust-rich samples).
The second part of a potential M1 process, the adsorption of the microbially produced excess
alkanes onto dust particles in the ice and the subsequent desorption during extraction, remains
difficult to evaluate. In particular why should in situ produced alkanes be adsorbed onto mineral
dust particles but not the atmospheric $CH_4$ that is anyway available in the air bubbles in the ice?
Apart from these quantitative limitations of microbial $CH_4$ in situ production in ice, there is
contradicting evidence from the "microbial inhibition experiment" by Lee et al. (2020) also for
microbial production of alkanes during extraction. Lee et al. (2020) tested whether biological
$CH_{4(xs)}$ production in the melt water was inhibited when the ice core samples were treated with
$HgCl_2$. As $CH_{4(xs)}$ was still observed in the poisoned samples and as it seems quite unlikely that
microbes are resistant to $HgCl_2$, this experiment questions the hypothesis of microbially
produced $CH_{4(xs)}$ also during extraction (*in extractu*).

At this point, a microbial production process seems rather unlikely but is not definitively ruled
out. However, our ratios of excess methane/ ethane/ propane in NGRIP and GRIP samples add
another piece of corroborating evidence that excess alkanes are not produced microbially. The
main microbial production process of methane, the decomposition of organic precursors in an
anaerobic environment by archaea, also co-produces ethane and propane, however only in
marginal amounts. The typical methanogenesis yields >200 times more methane than ethane
and propane (Bernard et al., 1977; Milkov and Etiope, 2018) while we find a molar ratio of
methane to ethane to propane of 14:2:1 in our samples. This renders a microbial production
pathway (in situ and *in extractu,* i.e. M1 and M2) for excess alkanes unlikely. Moreover, a
microbial production of $CH_4$ is unlikely in view of the $\delta^{13}C$-$CH_{4(xs)}$ signature which is too heavy
for microbial $CH_4$.

We conclude that regardless of the production pathway, in situ or *in extractu*, the fingerprint of
the produced excess alkanes in our samples (heavy $\delta^{13}C$-$CH_{4(xs)}$ signature and low
$CH_4/(C_2H_6+C_3H_8)$ ratio) essentially rules out a microbial source and another (abiotic?) process
for excess alkane production is likely to exist (see Fig. 11).

**(3) Abiotic/ chemical production**
In this last section we consider an abiotic or chemical process to be responsible for the observed
excess alkanes, where excess alkanes would be produced through the abiotic decomposition of



labile organic compounds in the melt water (C2). Again, we disregard an abiotic in situ
production in the ice (C1) based on the same arguments presented in the previous section for a
microbial in situ production, as it would require the quantitative adsorption of the in situ
produced alkanes onto mineral dust particles but not the atmospheric $CH_4$ that is available in
the ice otherwise.

Organic precursors for this abiotic production during extraction could be any organic matter
(either microbial or plant-derived). As the amount of excess alkanes is tightly coupled to the
amount of dust, we assume that these organic compounds are attached to dust particles. This
"docking" of the organic precursor onto the mineral dust could happen already in the dust
source region involving organic material available at the surface (East Asian deserts). Or by
adhering to volatile organic molecules or secondary organic aerosols from the atmosphere,
either before deflation at the source region or during transport to Greenland. Note that organic
substances might potentially experience abiotic preconditioning (ageing) during aerosol
transport and only the final step of alkane production may occur during the wet extraction.

We consider this pathway, as in recent years the prevailing paradigm that methane is only
produced by methanogenic archaea under strictly anaerobic conditions has been challenged.
Several experimental studies demonstrated that methane can also be released from dried soils
(Hurkuck et al., 2012; Jugold et al, 2012; Wang et al., 2013; Gu et al., 2016), fresh plant matter
and dry leaf litter (Keppler et al., 2006; Vigano et al., 2008, 2009, 2010; Bruhn et al., 2009;
Derendorp et al., 2010, 2011), different kinds of living eukaryotes (plants, animals and fungi)
(Liu et al., 2015), single organic structural components (McLeod et al., 2008; Messenger et al.,
2009; Althoff et al., 2014) and in fact under aerobic conditions. Most of these studies focused
on methane, however, there is also evidence for simultaneous formation of other short-chain
hydrocarbons like ethane and propane (McLeod et al. 2008; Derendorp et al., 2010, 2011). At
least three mechanisms have been identified to be relevant: i) photo-degradation, ii) thermal
degradation, or iii) degradation by the reaction with a reactive oxygen species (ROS) (Schade
et al., 1999; Wang et al., 2017). Common to all three pathways is a functional group (for
example a methyl or ethyl group) that is cleaved from the organic precursor molecule. Key
parameters that control the production of abiotic methane are mainly temperature, UV radiation,
water/ moisture, and the type of organic precursor material (Vigano et al., 2008; Derendorp et
al., 2010, 2011; Hurkuck et al., 2012; Jugold et al., 2012; Wang et al., 2013, 2017). This "new"
abiotic pathway of methane formation has not been discussed yet to be active during ice core



analyses, however, we believe that this process could be active during our melt extraction. In
the following section we discuss the key parameters that generally influence abiotic production
with respect to our measurement conditions and review the viability of this process for ice core
samples and in the light of our experimental observations.

Recent findings demonstrated the large variety of potential organic precursors for abiotic trace
gas formation. In general, the functional group cleaved from the precursor molecule defines the
species to be produced, thus methyl- (or ethyl-) group containing substances for the production
of methane (or ethane). For the formation of methane, the plant structural components pectin
and lignin have been identified in many studies as a precursor in different plant materials. Pectin
and lignin contain methoxyl-groups in two different chemical types, ester methoxyl (present in
pectin) and ether methoxyl (present in lignin) (Keppler et al., 2006, 2008; McLeod et al., 2008;
Messenger et al., 2009; Bruhn et al., 2009; Vigano et al., 2008; Hurkuck et al., 2012; Liu et al.,
2015; Wang et al. 2017). Ester methyl groups of pectin were also discovered as precursor for
ethane formation (McLeod et al., 2008). Overall, pectin makes up a large fraction of the primary
cell wall mass of many plants, thus, representing a large reservoir available as organic precursor
for abiotic alkane formation (Keppler et al., 2006; Mohnen et al., 2008; Vigano et al., 2008,
2010; McLeod et al., 2008), and may be present in sufficient quantities in our ice core samples
attached to mineral dust particles. $CH_4$ production was also detected from cellulose even though
it does not contain methoxyl groups suggesting that other carbon moieties of polysaccharides
might allow abiotic $CH_4$ formation (Keppler et al., 2006; Vigano et al., 2008). In addition, poly-
unsaturated fatty acids in plant membranes are suggested to play a key role not only in the
formation of methane but also for ethane and propane (John and Curtis, 1977; Dumelin and
Tappel, 1977; Derendorp et al., 2010, 2011). Further, sulfur-bound methyl groups of
methionine are an important precursor for abiotic $CH_4$ formation in fungi (Althoff et al., 2014).

Considerably different emission rates were found for the same amount but different type of
organic substances leading to the conclusion that abiotic emissions are strongly dependent on
the type of organic precursor material or single structural components (Keppler et al., 2006;
McLeod et al., 2008; Vigano et al., 2008; Messenger et al., 2009; Hurkuck et al., 2012). Other
factors such as leaf and cell wall structure (McLeod and Newsham, 2007; Watanabe et al., 2012;
Liu et al., 2015) and the organic carbon content (Hurkuck et al., 2012) are suggested to have an
important influence on this process, too.



To explain the observed excess alkanes in dust-rich Greenland ice core samples by an abiotic
production through the decomposition of labile organic compounds requires adequate quantities
of organic precursors to be present within the ice core samples. Certainly, such material is
present in Greenland ice, but currently, there is no record on the amount and type of organic
substances in NGRIP and GRIP ice available. We have some limited information from
occasional Greenland ice core samples in which different types of organic substances were
detected (Giorio et al., 2018, and references therein), but it does not allow for an overarching
interpretation for our ice samples. A NGRIP record on formaldehyde and a GRIP record on
acetate and formate exists (Fuhrer et al., 1997), but as these substances are only representative
for the respective dissolved organic compounds in the ice and not for any organic molecules
attached onto the dust particles, they show lower levels during the glacial.

However, we also have to question whether a perfect record of eligible precursor molecules
could at all exist. As we observe that precursor substances are labile and quickly decompose
when in contact with liquid water, a direct measurement of these substances might not be
possible but only for similar, non-reactive substances, which are then not qualified as precursors
for the reaction observed. The problems of sampling, analysis and interpretation of organic
material in polar ice are well summarized and expounded in Giorio et al. (2018).

In any case, it appears likely that the mineral dust, primarily coming from the Taklamakan and
Gobi deserts (Biscaye et al., 1997; Bory et al., 2003), carries along soil organic matter or plant
residues or accumulates organic aerosols as a result of organic aerosol aging during transport.
In our data we see a relationship between the amount of mineral dust within the ice core samples
and the amount of excess alkanes. As the amount of excess alkanes per $Ca^{2+}$ (or mass of dust)
is variable, this suggests that mineral dust is just a carrier for (a variable amount of) organic
substances but does not account for the production of excess alkanes itself. The dust content
within the ice core sample can therefore only serve as a rough estimate of organic precursor
availability and whether an abiotic production from organic precursor substances is likely to
occur during extraction.

Again, our experiments can shed some light on the viability of this pathway for excess alkane
production. If we assume that the dust-related organic matter in the ice represents a reservoir
available for an abiotic production, then the decomposition continues until all functional groups
are cleaved from their organic precursor molecules and released as excess alkanes. Once the



reservoir is emptied excess alkane production ceases (Derendorp et al., 2010, 2011). In line, we
interpret that the decrease in the amount of measured excess alkanes from the 1$^{st}$ to the 2$^{nd}$
extraction may result from an exhaustion of the precursor reservoir. The reaction time is slow
enough to allow for the continuing production during the second extraction but too slow for a
detectable production during continuous flow analysis of CH$_4$, where the water phase is present
only for less than a minute before gas extraction. The significantly reduced production during
the 2$^{nd}$ extraction in our samples shows that the time scale for this process is hours (see Fig.
B1) until the reservoir of functional groups is depleted. We note that this implies that the amount
of excess alkanes is strongly dependent on the time span when liquid water is in contact with
the dust, which varies among the methods used for CH$_4$ analyses. Thus, any excess CH$_4$ in
measurements from different labs performed under different conditions may differ.

To explain an abiotic alkane production, certain conducive boundary conditions must be met.
The most important parameters that control non-microbial trace gas formation are temperature
and UV radiation. This was demonstrated in many field and laboratory experiments (Keppler
et al., 2006; McLeod et al., 2008; Vigano et al., 2008, 2009; Messenger et al., 2009; Bruhn et
al., 2009; Derendorp et al., 2010, 2011; Hurkuck et al., 2012; Jugold et al., 2012; Wang et al.,
2017). Generally, increasing temperatures lead to exponentially increasing CH$_4$ emissions
(Vigano et al., 2008; Bruhn et al., 2009; Wang et al., 2013; Liu et al., 2015). The same behaviour
was observed for ethane and propane with very low emissions at ambient temperatures (20-
30°C) and a maximum at 70°C (McLeod et al., 2008; Derendorp et al., 2010, 2011). At constant
temperatures emission rates decreased over time, which is at high temperatures on the timescale
of hours and at ambient temperatures of months. Even after months, some production was
observed, pointing to a slowly depleting reservoir of organic precursors (Derendorp et al., 2010,
2011). Increasing emissions observed at temperatures >40°C were also used as indicator to
exclude the possibility of enzymatic activity, as the denaturation of enzymes would lead to
rapidly declining emissions at higher temperatures (Keppler et al., 2006; Derendorp et al., 2011;
Liu et al., 2015). We note that our sample extraction takes place at 0°C or a few °C above,
hence, temperature conditions during the extraction are not conducive of the type of abiotic
alkane production as observed in the studies listed above. Whether the cool temperature of the
melt water during extraction inhibits abiotic reaction is difficult to conclude. Derendorp et al.
(2010, 2011) observed a much lower temperature dependency of C2-C5 hydrocarbon emissions
from ground leaves than whole leaves, which might also apply to our samples with very fine
fragments of organic substances attached to dust particles.




Besides the strong relationship to temperature also UV irradiation seems to have a substantial
effect on an abiotic production. Studies on irradiated samples (dry and fresh plant matter, plant
structural components) showed a linear increase in methane emissions, while UV-B irradiation
seems to have a much stronger effect on the release compared to UV-A (Vigano et al., 2008;
McLeod et al., 2008; Bruhn et al. 2009; Jugold et al., 2012). The influence of visible light (400-
700 nm), however, seems controversial (Keppler et al., 2006; Bruhn et al., 2009; Austin et al.,
2016). Further, samples that were heated and irradiated show a different emission curve than
just heated samples, indicating that irradiation changes the temperature dependency, in turn
pointing to the fact that different chemical pathways exist (Vigano et al., 2008).
In dark experiments on plant material at different temperatures $CH_4$ emissions were still
observed, while again higher temperatures revealed much higher emissions, emphasizing the
strong temperature dependency also without UV irradiation (Vigano et al., 2008; Wang et al.,
2008; Bruhn et al., 2009). The release of ethane along with methane from pectin was also
stimulated under UV radiation (McLeod at al., 2008).

Regarding our measurements, the sample vessel in the $\delta^{13}C$-$CH_4$ device is encased by a UV
blocker foil absorbing the shortwave (<600 nm) emissions from the heating bulbs when melting
the ice sample, while in the $\delta D$-$CH_4$ device the sample vessel is completely shielded from light
(Sect. 2.2 and 2.3). Two NGRIP ice core samples were measured with the $\delta^{13}C$-$CH_4$ device in
the dark ("dark extraction") showing the same amount of excess alkanes as the regular
measurements at day light. This indicates that light >600 nm has no influence on an *in extractu*
reaction during our measurements. We stress that although we can exclude a direct UV effect
during sample extraction, it is possible that UV irradiation during dust aerosol transport to
Greenland and within the upper snow layer after deposition until the snow gets buried into
deeper layers may precondition organic precursors attached to mineral dust to allow for alkane
production to occur during extraction. In particular, the first step of the reaction (excitation of
the homolytic bond of a precursor compound) may start already in the atmosphere or in the
upper firn layer where energy from UV radiation is available. Within the ice sheet the reaction
may be paused and only becomes reactivated during the melting process when liquid water is
present.

Finally, we consider the role of reactive oxygen species in an abiotic production pathway. ROS
are widely produced in metabolic pathways during biological activity but also during



photochemical reactions with mineral oxides (Apel and Hirt, 2004; Messenger et al., 2009;
Georgiou et al., 2015). Through their high oxidative potential ROS are capable to cleave
functional groups from precursor compounds. Several studies have demonstrated this
mechanism for the production of abiotic $CH_4$ in soils and plant matter (McLeod et al., 2008;
Messenger et al., 2009; Althoff et al., 2010, 2014; Jugold et al., 2012; Wang et al., 2011, 2013)
and for other trace gases such as $CO_2$, ethane, and ethylene from plant pectins (McLeod et al.,
2008). UV radiation or thermal energy has no direct influence on the degradation process by
the reaction with ROS, however, it might also be a stimulating factor and evoke further indirect
reactions. For instance, UV radiation can lead to changes in plants which in turn lead to ROS
generation (Liu et al., 2015). It was demonstrated that UV radiation induces the formation of
organic photosensitizers or photo-catalysts which increase $CH_4$ emissions from pectin
(Messenger et al., 2009) and clay minerals. For example, the formation of OH from
montmorillonite and other clay minerals upon UV (and visible light) irradiation shows that
clays might play a significant role in the oxidation of organic compounds on their surface in
different environments (Katagi, 1990; Wu et al., 2008; Kibanova et al., 2011).

It has been proven that the species type and the overall amount of ROS available for, or involved
in a reaction, has a significant effect on the amount of emissions through such a process (Jugold
et al., 2012; Wang et al., 2013, 2017). For the production of methane (and ethane), hydrogen
peroxide ($H_2O_2$) and hydroxyl radicals (OH) have been proven to be the prominent species
(Messenger et al., 2009; Althoff et al., 2010; Wang et al., 2011; Jugold et al., 2012; Wang et
al., 2013, McLeod et al. 2008). Such ROS could be already present in the snow and ice or being
produced in the melt water. For example, $H_2O_2$ can be unambiguously detected in Greenland
Holocene ice using CFA, however, $H_2O_2$ in dusty glacial ice is mostly below the detection limit,
likely due to oxidation reactions in the ice sheet or during melt extraction.

In summary, we believe that in our case of excess alkane production/ release in the melt water
at low temperatures and without any UV irradiation the ROS-induced mechanism appears
possible. In experiments with plant pectin McLeod et al. (2008) observed not only $CH_4$ but also
ethane and found a methane to ethane production ratio of around 5 which is similar to our value
of around 7. Accordingly, we see that a ROS-induced production pathway has the potential to
explain excess alkanes in our samples, however, little is known about ROS chemistry in ice
cores analyses in particular for reactions with organic precursors and more research is needed
to understand the role of ROS in organic decomposition in ice.

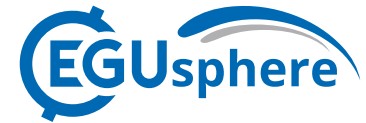

Another key parameter influencing all abiotic pathways might be the presence of liquid water
or moisture. In experiments testing the hypothesis of non-microbial $CH_4$ formation in different
soil samples, it was demonstrated that the addition of water/ moisture led to an up to 8-fold
increase in $CH_4$ emissions (Hurkuck et al., 2012; Jugold et al., 2012; Wang et al., 2013). It is
hypothesized that the presence of liquid water or moisture stimulates (in addition to heating or
UV radiation) the cleaving process of a functional group from the primary precursor compound
and therefore increases the production of $CH_4$. However, it seems that the stimulating effect by
water cannot be generalized, as Wang et al. (2013) emphasized that this process is highly
dependent on "water of proper amount". In their experiments, $CH_4$ emissions from peat and
grassland soil samples treated with a varying amount of water in oxia–anoxia cycles at 70°C
were measured. They observed that under both aerobic and anaerobic conditions water does not
always stimulate non-microbial $CH_4$ release and that too much water can also suppress $CH_4$
emissions. As Hurkuck et al. (2012) and Jugold et al. (2012) only observed a positive effect of
water on $CH_4$ emissions in oxic soils, it is hypothesized that the amount of water they added to
their samples is by chance in the stimulating range (Wang et al., 2013). In addition, Wang et al.
(2013) observed differences between different soil samples in response to a varying water
content indicating that the water effect is different for different precursors. With respect to our
observations on NGRIP and GRIP samples the presence of water seems to be a fundamental
parameter influencing an *in extractu* process, where the duration of water presence plays an
important role in these reactions.
A final puzzle piece for a possible abiotic methane production comes from our dual isotopic
fingerprints of the excess $CH_4$. As illustrated in Fig. 11 both $\delta^{13}C$ and $\delta D$ of the $CH_4$ produced
are in overall agreement with the carbon and hydrogen isotopic composition of potential organic
precursors. For $\delta^{13}C$ our values lie on the heavier side of the isotopic carbon signature spectrum
but still within the wide distribution of possible isotopic precursor signatures, for $\delta D$ the
signature lies well within the distribution.

We conclude that despite our inability to pinpoint the exact organic precursors that lead to
abiotic excess alkane production during the melt extraction of our ice samples at this point, both
the ratio of the excess alkanes as well as the isotopic signature of excess $CH_4$ is generally in
line with this pathway. Thus, without further contradicting evidence from targeted studies on
organic precursors in ice core samples and their degradation by ROS, we believe that the ROS-
induced production pathway is the most likely explanation for the observed excess alkane
production during extraction.



**5. Conclusions and Outlook**

The comparison of methane records from ice cores samples measured with different extraction techniques requires careful consideration and interpretation. Non-atmospheric methane contributions to the total methane concentration were discovered in specific Greenland ice core sections pointing to a process occurring during the wet extraction. To better assess this finding, we measured new records of [methane], [ethane], [propane], $\delta$D-CH$_4$, and $\delta^{13}$C-CH$_4$ on discrete NGRIP and GRIP ice core samples using two different wet extraction systems. With our new data we confirm the production of CH$_{4(xs)}$ in the melt water and quantify its dual isotopic signature. With the simultaneous detection of ethane and propane we discovered that these short-chain alkanes are co-produced in a fixed ratio pointing to a common production pathway. With our 2$^{nd}$ extraction we constrained the temporal dynamics of this process, which occurs on the timescale of an hour.

Based on our new experimental data we provide an improved assessment of several potential mechanisms that could be relevant for the observed variations in NGRIP and GRIP ice samples. A microbial CH$_4$ production represents an obvious candidate but regardless of whether this CH$_4$ is produced in situ or *in extractu,* several lines of evidence gained from our measurements (low CH$_4$/(C$_2$H$_6$+C$_3$H$_8$) ratio, heavy $\delta^{13}$C-CH$_{4(xs)}$ signature) demonstrate that the fingerprint of the produced excess alkanes is unlikely to have a microbial source. Also an adsorption-desorption process of atmospheric or thermogenic CH$_4$ on dust particles does not match many of our observations (low CH$_4$/(C$_2$H$_6$+C$_3$H$_8$) ratio, light $\delta$D-CH$_{4(xs)}$ signature) and is therefore unlikely. However, with the current knowledge we cannot definitely exclude such a process to be responsible for the observed excess alkane levels in our samples.

At present we favor to explain the formation of excess alkanes by abiotic decomposition of organic precursors during prolonged wet extraction. Such an abiotic source for methane and other short-chain alkanes was discovered previously in other studies (Keppler et al., 2006; Vigano et al., 2008, 2009, 2010; Messenger et al., 2009; Hurkuck et al., 2012; Wang et al., 2013, and others listed above) using different organic samples, e.g. from plant or soil material, however, this process has not been connected to excess CH$_4$ production in ice core analyses. This process matches many of our observations and such a mechanism can be responsible for excess alkanes in Greenland ice core samples. To better assess a potential abiotic production process in ice analyses the most important questions to solve in the future are: What are the specific precursor substances? Which parameters control an abiotic production during wet



extractions? How does the fixed molar ratio between methane, ethane, and propane come about
in this process? And finally, in which way is this excess alkane production causally related to
the amount of mineral dust within the ice sample?

Identifying a specific reaction pathway that leads to the short-chain alkanes with their observed
ratios would certainly benefit from identifying targeted organic precursor substances in the ice.
However, detecting these postulated organic precursors in the ice core is inherently difficult as
these compounds must be very labile in water as our experiments demonstrated that after about
30 min only a fraction of these compounds remains in the melt water while the majority already
reacted to excess alkanes. Future studies may also focus on further isotopic measurements
($\delta^{13}$C-CH$_4$ and $\delta$D-CH$_4$) including isotope labeling experiments providing an option to
unambiguously detect methane produced during the measurement procedure in a commonly
used wet extraction technique, and again, to uncover potential reaction mechanisms for CH$_{4(xs)}$
production.

To better assess the viability of the alternative hypothesis of a release of previously adsorbed
alkanes from dust particles (scenario A1 and A2) during the extraction, dust particles from the
Taklamakan or Gobi desert need to be tested whether they contain relevant amounts of adsorbed
alkanes that are released when in contact with liquid water. A second step could be to expose
such dust samples to high levels of alkanes to mimic the adsorption process of natural gas seeps.
It also needs to be shown that the adsorbed alkanes stay adsorbed on the dust particles for a
prolonged time (months, ideally years) after exposing the particles to ambient air and that
droplet and ice nucleation during aerosol transport does not lead to a loss of the previously
adsorbed CH$_4$. To quantify any isotopic fractionation involved with the ad- and desorption step,
$\delta^{13}$C-CH$_4$ and $\delta$D-CH$_4$ analyses will be most valuable.

Finally, our studies clearly show that the published Greenland ice core CH$_4$ record is biased
high for selected (glacial) time intervals and needs to be corrected for the excess CH$_4$
contribution. This is particularly important for studies of the IPD in CH$_4$ and stable isotope
ratios of methane. Methodological ways to remedy excess methane (and ethane and propane)
in future measurements of atmospheric [CH$_4$] from air trapped in ice cores could be to use
continuous online CH$_4$ measurements, which apparently avoid CH$_{4(xs)}$ production. But also dry
extraction methods and sublimation techniques for discrete samples, which are expected to
avoid *in extractu* production by evading the melting phase, could be used. Finally, our own





$\delta^{13}$C-CH$_4$ device, which allows to measure $\delta^{13}$C-CH$_4$ as well as methane, ethane, and propane
concentrations from the same sample, can be used to correct the measured CH$_4$ values making
use of the co-production of the other two alkanes.

It is clear that CH$_{4(xs)}$ needs to be corrected for when interpreting the already existing discrete
CH$_4$ records and its stable isotopes in dust-rich intervals in Greenland ice core samples. Impact
of CH$_{4(xs)}$ on interpreting past atmospheric [CH$_4$] will only slightly affect radiative forcing
reconstructions, however, it will have a significant effect on the assessment of the global CH$_4$
cycle and in particular on the hemispheric CH$_4$ source distribution which is based on the IPD.
We observe that in some intervals CH$_{4(xs)}$ is in the same range as the previously reconstructed
IPD implying that correcting for CH$_{4(xs)}$ will lower the IPD considerably and hence lower also
the relative contribution of northern hemispheric sources at those times. We see that there is the
urgent need to reliably revisit Greenland ice core CH$_4$ records for the excess CH$_4$ contribution
and in future work we aim to establish an applicable correction for excess methane (CH$_{4(xs)}$,
$\delta^{13}$C-CH$_{4(xs)}$, $\delta$D-CH$_{4(xs)}$) in existing records using the co-production ratios of methane, ethane,
and propane, the isotopic mass balance of excess and atmospheric CH$_4$ in ice core samples as
well as the overall correlation of excess CH$_4$ with the mineral dust content in the ice.




















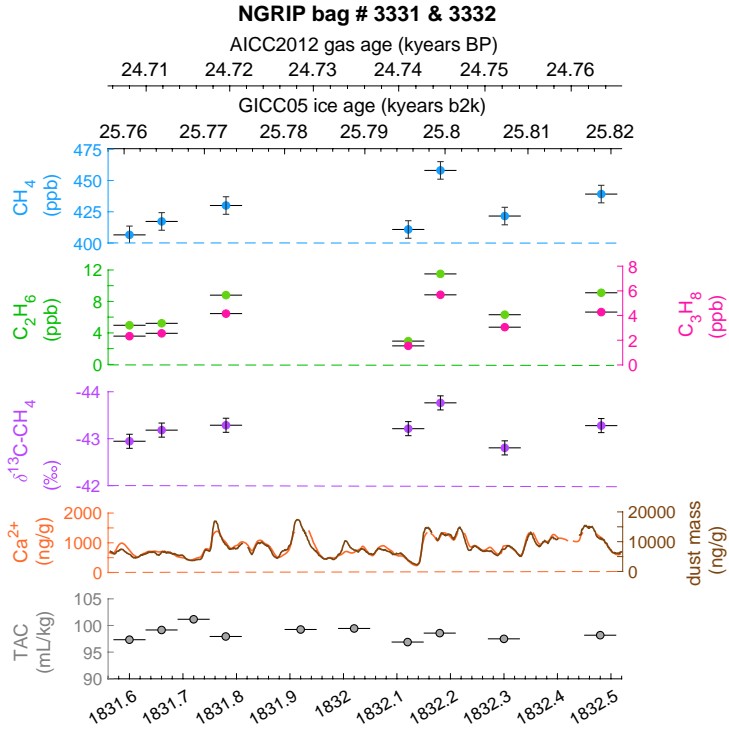


**Figure A1: Detailed data overview for the neighbouring NGRIP bags 3331 & 3332.** Bag-specific overview of
several parameters measured for each sample in this bag: methane, ethane, propane, $Ca^{2+}$, mineral dust mass, TAC
(Total Air Content), $\delta^{13}C$-$CH_4$, indicated at the NGRIP depth (bottom axis) and the AICC2012 gas age (upper top
axis) and the GICC05 ice age (lower top axis). The mineral dust record is taken from Ruth et al. (2003), the $Ca^{2+}$
record from Erhardt et al. (2022).

1273

1274



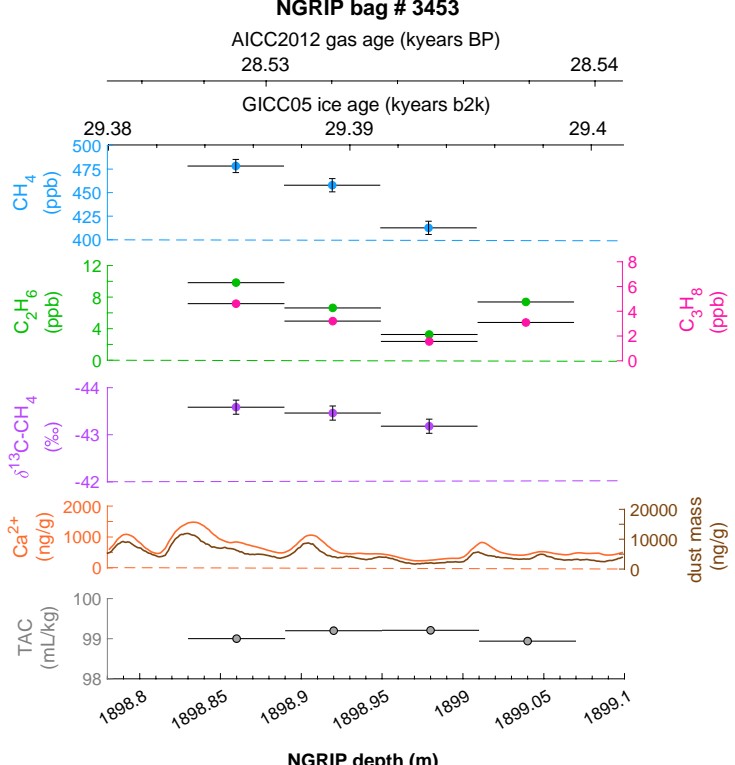

**Figure A2: Detailed data overview for NGRIP bag 3453.** Bag-specific overview of parameters measured for each sample in this bag: methane, ethane, propane, $Ca^{2+}$, mineral dust mass, TAC (Total Air Content), $\delta^{13}$C-CH$_4$, indicated at the NGRIP depth (bottom axis) and the AICC2012 gas age (upper top axis) and the GICC05 ice age (lower top axis). The mineral dust record is taken from Ruth et al. (2003), the $Ca^{2+}$ record from Erhardt et al. (2022).





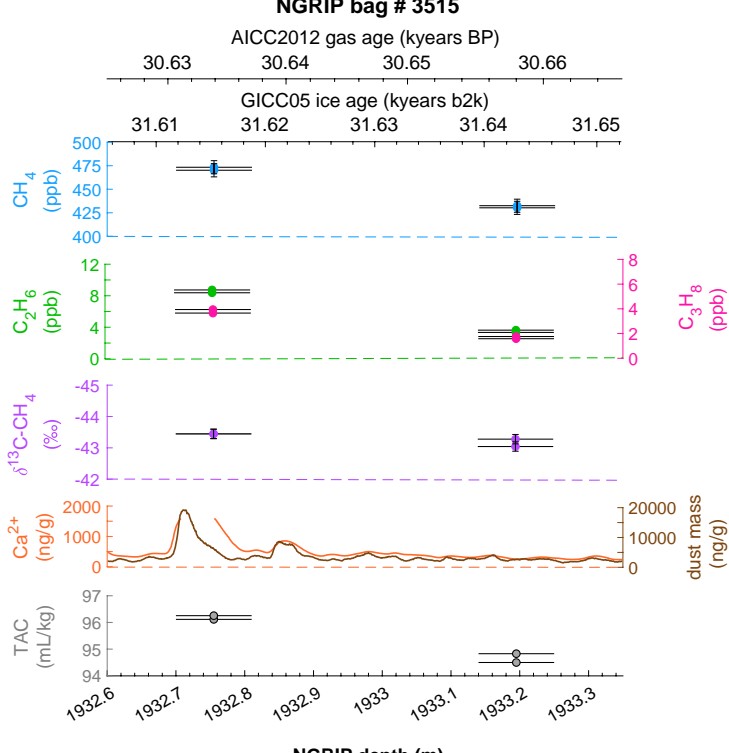

**Figure A3: Detailed data overview for NGRIP bag 3515.** Bag-specific overview of parameters measured for
each sample in this bag: methane, ethane, propane, $Ca^{2+}$, mineral dust mass, TAC (Total Air Content), $\delta^{13}C$-$CH_4$,
indicated at the NGRIP depth (bottom axis) and the AICC2012 gas age (upper top axis) and the GICC05 ice age
(lower top axis). The mineral dust record is taken from Ruth et al. (2003), the $Ca^{2+}$ record from Erhardt et al.
(2022). Note that there is a gap in the $Ca^{2+}$ record which was corrected by a fill routine for the analysis of the two
measured samples at this depth.













**Appendix B**

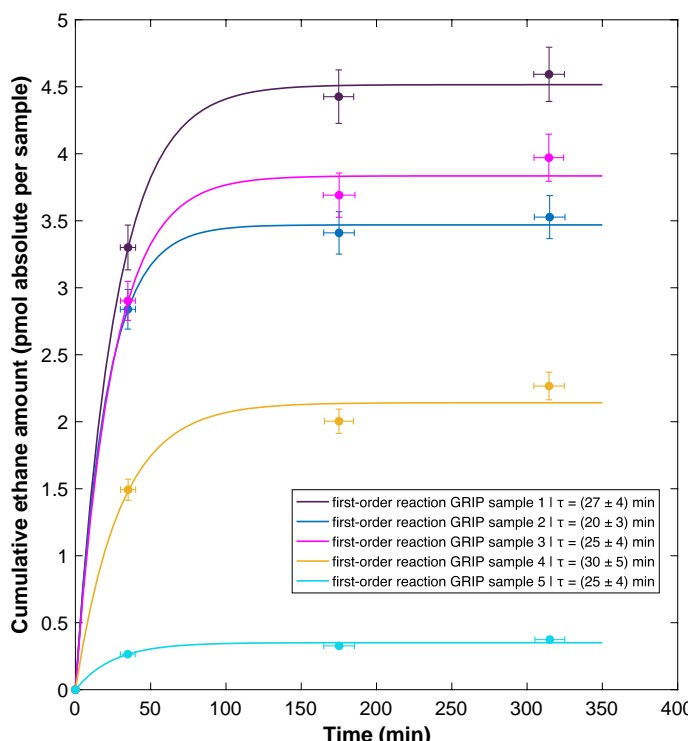


**Figure B1: Temporal dynamics of excess ethane production in GRIP ice core samples.** Cumulative ethane
amount from the 1st, 2nd, and 3rd extraction in relation to the time available for a potential reaction in the melt water
during each extraction. We assume a first-order reaction kinetic as model for our observations where the mean
half-life time ($\tau$) and standard deviations are calculated for each GRIP sample from the compilation of all 1000
iterations of our Monte Carlo approach assuming an uncertainty in x of ± 5 min and an uncertainty in y of ± 5 %
of the measured value in the 1st extraction and ± 10 % in both the 2nd and 3rd extraction.














**Author contribution**

MM and BS performed the measurements; MM and JS analyzed the data; MM wrote the manuscript draft; MM prepared the manuscript with contributions from all co-authors.

**Competing interests**

The authors declare that they have no conflict of interest.

**Acknowledgments**

The research leading to these results has received funding from the Swiss National Science Foundation (no. 200020_172506 & 200020B_200328). This work is a contribution to the NorthGRIP ice core project, which is directed and organized by the Department of Geophysics at the Niels Bohr Institute for Astronomy, Physics and Geophysics, University of Copenhagen. It is supported by funding agencies in Denmark (SNF), Belgium (FNRS-CFB), France (IFRTP and NSU/CNRS), Germany (AWI), Iceland (RannIs), Japan (MEXT), Sweden (SPRS), Switzerland (SNF), and the United States (NSF).




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
