# Peer review of "Methane, ethane, and propane production in Greenland ice core samples and a first isotopic characterization of excess"

_EGUsphere, 2022_

## Author Comment (AC1)

Mühl et al. present compelling evidence that there is production of methane, ethane, and propane during the wet (melt) extraction of air from Greenland ice cores with high dust content. This paper builds on the previous work by Lee et al. (2020) who showed pretty convincing evidence to the same effect. This paper expands previous work by presenting new measurements of methane and its stable isotopes (13C and D), as well as ethane and propane. Their results confirm the findings of Lee at al. (2020), and primarily via use of the new isotope data, they elaborate in detail on the possible mechanisms that may be responsible for the excess light alkanes observed in dusty Greenland ice. The results have important implications in that the existing ice core methane records from Greenland, particularly those that are based on discrete measurements, likely include some excess methane during periods that correspond to the stadial ice form the last glacial with high dust concentrations. The presence of excess methane also significantly impacts the stable isotope records methane.

Thank you Murat Aydin for this extensive and thoughtful review.

The ice core air extraction and measurement methods have been established previously and the data are clearly presented in most cases (some exceptions listed below). I have one major disagreement with the findings and conclusions offered in this paper. The authors single out an abiotic production mechanism driven by reactive oxygen species (ROS) as the likely cause of the excess alkanes in their extraction vessels. This argument rests primarily on the isotopic evidence, which in their assessment points strongly in the ROS mechanism direction. In my evaluation, the isotope data, along with the alkane ratios, are just as consistent with the adsorption/desorption mechanism. The possible explanations offered for how an abiotic process might work via the ROS mechanism are not very convincing (details below). The discussions and conclusions of the paper regarding the possible mechanism should be more balanced between the adsorption/desorption vs. the abiotic production mechanisms.
We rephrased this part of the abstract, so that it becomes more clear that we do not rule out an adsorption/desorption process of thermogenic gas. It should now be more balanced. The same applies for the discussion section.

Changes in the text:

*... With the co-production ratios of excess alkanes and the isotopic composition of excess methane we established a fingerprint that allows us to constrain potential formation processes. This fingerprint is not in line with a microbial origin. Moreover, an adsorption-desorption process of thermogenic gas on dust particles transported to Greenland appears also unlikely. Rather the alkane pattern appears to be indicative of abiotic decomposition of organic matter as found in soils and plant leaves.*

*... In contrast, our hydrogen isotopic measurements on NGRIP samples reveal a very light $\delta D$-$CH_{4(xs)}$ value (Keeling y-intercept weighted mean) of (-326 ± 57) ‰ and slightly outside of the field of a thermogenic origin (see Fig. 11). The value is similar to the estimate by Lee at al. (2020), which, however, lies inside the field of a thermogenic origin (see Fig. 11). While both the low $CH_4/(C_2H_6+C_3H_8)$ ratio and the $\delta^{13}C$-$CH_{4(xs)}$ could be indicative of a thermogenic source (A1), the light $\delta D$-$CH_{4(xs)}$ signature is far away from the atmospheric $\delta D$-$CH_4$ value and is borderline in line with typical $\delta D$-$CH_4$ values of a thermogenic origin. Hence, our $\delta D$-$CH_{4(xs)}$ values exclude the atmospheric adsorption scenario A2 and put a question mark after the seep adsorption scenario A1. However, for the seep adsorption scenario A1 to work the dust particles on which the thermogenic gas adsorbed are not allowed to experience any contact with liquid water prior to the analysis in the lab. In other words, if the particles get in contact*

*with liquid water after the adsorption step, the adsorbed alkanes would desorb from the particles as they do it in the laboratory during melting. Given the occurrence of wet/dry cycles in the source area (Ruth et al., 2007), we question the plausibility of scenario A1.*

*... Thus, without further contradicting evidence from targeted studies on organic precursors in ice core samples and their chemical degradation, we believe that the ROS-induced production pathway is to date the most likely explanation for the observed excess alkanes during extraction. However, we cannot completely rule out an adsorption-desorption process of thermogenic gas on dust particles.*

*... Also an adsorption-desorption process of atmospheric or thermogenic $CH_4$ on dust particles does not match many of our observations and is therefore unlikely. However, with the current knowledge we cannot definitely exclude such an adsorption of thermogenic gas to be responsible for the observed excess alkane levels in our samples.*

Structurally, the paper is well organized, although some of the later discussion sections (possible mechanisms) are unnecessarily lengthy, containing repetitive or not directly relevant information. An overall editing effort would be beneficial. I offered some suggestions in line-by-line comments but more can be done.

We shortened the text wherever possible in our revisions (see track changes in the reviewed manuscript.

**Comments on the interpretation of the isotope data**

I'm concerned about the determination of the $^{13}CH_4$ value that is best representative of the $xsCH_4$ from the measurements. There are two sets of results. The weighted mean from the Keeling plots suggests a lighter isotopic signature than what is measured whereas the 2nd extraction results suggest a heavier one. You argue that due to the uncertainties, the results from the two methods cannot be identified as different (how did you test this, the 2sigma uncertainties you report suggest they are different), then go on use the Keeling plot result as the representative value and ignore the measurements of the 2nd extraction methane isotope ratio.

Sorry for the confusion and thank you for this important notice! The uncertainties given in the numbers are 1 sigma here and throughout the manuscript. We changed it accordingly. This implies, that the results are not significantly different within the 2 sigma error, as stated in our manuscript.

I do not think you have good justification to ignore the 2nd extraction results. You have an adequate number of measurements to characterize the true mean here. The larger uncertainty of a single 2nd extraction measurement should not be a real concern since it matches the 2 SD (is this 2 SD or 2 s.e., explain briefly in the captions) of all your measurements, which is at the very least equivalent to the uncertainty of the estimate from the Keeling method. Further, your blank measurements are in good agreement with the 2nd extraction measurements, with no indication of bias. Therefore I see no reason to suspect the fractional contribution from the blanks is a significant issue either. In fact based on these measurements, one can argue that the methane in your blanks is from the same source as the methane in the 2nd extraction. Why could this be? More than one rection going on or fractionation during desorption are two possibilities that come to my mind.

Sorry again, it is 1 sigma here.

Reading your review comments, we also realized that to prevent ambiguities we need to better explain how we dealt with the blank contribution throughout the manuscript. Throughout the paper, we showed the data without blank correction, i.e. as measured which is the ice core derived amount plus the amount derived from the system (blank). Except for the $CH_4$ amount measured in the 2nd extraction which has a considerable "blank" contribution, the blank values for ethane and propane for both the 1st and the 2nd extraction are sufficiently small compared to the sample-derived amount. The advantage of showing values as measured is that we can plot both the blank and ice core values in a single figure which allows to see the size of the blank contribution and also the respective alkane ratio of the blank contribution.

With regard to $\delta^{13}C$-$CH_4$, indeed the $\delta^{13}C$-$CH_4$ signature of the blank (EDC) is similar and only a few ‰ heavier (-39.0 ‰) to the signature of our Greenland samples. Applying an isotope mass balance approach, we see that the leverage on our NGRIP values is small (0.31 ‰). Thus, applying a blank correction has only little influence and shifts the sample values a bit towards isotopically lighter values, and therefore more into the direction of the values obtained from the Keeling plot approach. For the sake of the length of the paper, we did not expand on these corrections for the $\delta^{13}C$-$CH_4$-signature. We wish to stress that measuring the $\delta^{13}C$-$CH_4$ signature of such small $CH_4$ amounts as available from the 2nd extraction is at the edge of what is possible with our device.

We clarified in the main text that we are not performing a blank correction (except for Fig. C1 where the blank correction is necessary).

We also clarified the section explaining the blank determination and precision of our method.

We added another Figure (Appendix B, Figure B1) with a description of the blank determination.

Figure 8 is revised.

[revised manuscript text omitted]

In contrast, the N (sample size) in the Keeling plots is low for all bags and you need an aggressive extrapolation to estimate the y intercept because most of the methane is atmospheric. This is a bad combination that lowers the confidence in the uncertainty estimates for the individual regressions.

That is of course correct but is included in the large uncertainties of the y-intercept determined with this Keeling approach.

At low N, the SD (or s.e.) of analytical linear regression parameters are biased, hence not reliable. It is not clear how the reported uncertainty of the weighted mean is calculated, but I think this estimate is just as uncertain because there are four bags and the uncertainty of each intercept estimate is not known well. Consequently, I suspect the true uncertainty in the Keeling plot approach could be higher than what is reported. The combined data from two neighboring bags (3331&3332) have higher N, but that bag spans more than 50 years (Fig. A1) so the constant atmospheric methane assumption is questionable and this combined bag points to the lightest intercept, which is clearly different than the 2nd extraction result.

We are aware that our measurements have uncertainty in both their x ($CH_4$ concentration) and their y values ($\delta^{13}C$-$CH_4$). For that reason we use the Yorkfit as it does not systematically bias the regression. We have also better explained this in the text now. See Hoheisel et al. (2019), Wehr and Saleska (2017), York (1968) and York et al. (2004) for more details.

Changes in the text:

*... Note that all regression lines are calculated by following the method of York (1968) and York et al. (2004). York's analytical solution to the best-fit line accounting for normally distributed errors both in x and y is widely used to determine an isotopic mixing line and has been proven as the least biased method (Wehr and Saleska, 2017; Hoheisel et al., 2019). Throughout the manuscript we use the 1 sigma (1 σ) standard deviation to express uncertainties.*

The neighboring bags 3331 & 3332 exactly span 50 years, which is close to the width of the gas age distribution at that time, thus any atmospheric variability should be substantially damped. Moreover, in this time period variations in the atmospheric $CH_4$ concentration are very low (samples are not within a DO event or a rapid $CH_4$ increase/ decrease as also known from Antarctic records not subject to excess methane production. However, if you have a look at Fig. A1, you can see that the 5[th] data points (from left) is clearly higher in methane, but also in ethane and propane, compared to the other data points in these 2 bags. This is due to a high *in extractu* production, but not due to a high atmospheric background. The two data points to the left and right of this high in extractu data point are lower, reflecting that this is no atmospheric variation as the age difference between the points is only 10 years which should be easily damped away by the gas age distribution.

Changes in the text:

*... Note that the two NGRIP bags 3331 and 3332 are neighbouring bags and were therefore combined into one Keeling y-intercept. As the individual samples in these two bags span less than 10 years between each other, they are the same within the age distribution, and the assumptions for the Keeling-plot approach (see Sec. 2.1) are met.*

Even assuming the currently reported uncertainties are an accurate estimate of the true uncertainty, I do not see a reason to pick results from one method over the other. A simple and conservative approach to solve this issue would be to accept the high uncertainty for these results. In my view, the possible range of the true 13C signature extends from the Keeling plot result to the 2nd extraction result. In any case based on these measurements, I agree that you can reject the microbial hypothesis (in the absence of fractionating processes), but it does not seem a high probability that the 13C source signature is within the C3 and C4 ranges shown in Fig. 11b.

We have changed Fig. 11. - instead of the pink star, there is now a pink dot with error bars for x and y. We agree with you that our $\delta^{13}C$-$CH_4$ results do not match the results from the Keppler and Vigano papers which have lighter $\delta^{13}C$ values that our ice core results.

Most of my above comments about the uncertainty in the Keeling plots are applicable to Fig. 9. You N is 4 and 5 respectively for the bags, and you only have 2 bags. You make a similar decision to the 13C case and ignore the Lee et al. (2020) estimate. It is quite possible that the D-CH4 signature can be -300 or higher, in which case you really cannot rule out a thermogenic source.

We agree with you that the $\delta D$-$CH_4$ signature can be higher than -300 permil as this is well within our 1 sigma uncertainty range. We do not rule out a thermogenic source based on the $\delta D$-$CH_4$ results alone, but also in the light of the overall circumstances. In particular, we also take into account the overall plausibility of the sequence of processes behind each of the discussed mechanisms. In case of the A1 mechanism "Thermogenic natural gas adsorption" we have to start with the adsorption of the alkanes on the mineral particles which afterwards

should not experience any contact with liquid water prior to the analysis in the lab. In other words, if the particles get in contact with liquid water after this adsorption step (either at the source or in the atmosphere), the adsorbed alkanes would desorb from the particles as they do it in the laboratory during melting, thus rendering A1 implausible even if all our isotopic and gas ratio fingerprints would match that mechanism.
Note that it is known that the deserts in the Tarim basin receive regular input from water from the surrounding mountain regions that also provide the minerals to the basin that are blown out of the desert afterwards.

**Changes in the text:**

*... This water contact could occur for example already at the dust source, as it is known that the deserts in the Tarim basin receive regular input from water from the surrounding mountain regions also providing the minerals to the basin that are blown out of the desert afterwards (Ruth et al., 2007).*

**Comments on the abiotic/chemical reactions and ROS**

The abiotic/chemical reactions from other natural systems that are discussed in detail mostly relate to terrestrial systems and are not good analogues in my opinion. The abundance of organic material is often very high and exposed to high temperatures and UV. Aquatic systems may be a better analogue (Li et al., 2022; Zhang and Xie, 2015), but both of these studies also show production is strictly tied to UV and report no methane production in the dark. I understand the desire to invoke ROS, but with no UV and reactions that go on for hours (1st, 2nd, 3rd extractions and probably beyond), and no $H_2O_2$ in the ice, I just cannot see which ROS could be generated in the vessel and how. Additionally, I was a little surprised to not see any consideration of surface reactions that can be catalyzed by transition metals. Metal particle concentrations in the ice tend to correlate with the dust, which may be the reason why there is a correlation between alkane production and dust in the first place. This is merely an idea, of course, in the absence of clear and direct evidence for reduction of complex organics in the dark leading to methane production. I'm not arguing that we can rule out this type of process, but any suggestion to this effect is quite speculative, and the language in the manuscript should reflect that.

As you pointed out the methane production in the two publications you suggested are strictly tied to UV radiation, which we do not have in our meltwater. We totally agree that our observations do not fit the classical ROS chemistry which typically require 4 ingredients simultaneously (UV, peroxides, organic material, liquid water). For that reason, we came up with the idea that our *in extractu* problem might be the result of a 2-stage reaction. In the 1st stage only 3 ingredients (UV, peroxides, organic matter) react in the absence of liquid water to produce an intermediate molecule (precursor) rather than $CH_4$ as the final product. The required ingredients for this 1st stage are matched both during transport in the high troposphere but also in the fresh upper snow layer known for its active UV-driven chemistry. The 2nd stage of the reaction only takes place when the intermediate molecule gets in contact with liquid water to break apart the precursor and release the alkanes as the final product of this 2-stage process. The key is that the 2nd stage does not require extra peroxides or UV, but just the presence of water. Since this "dark reaction" of the intermediate with liquid water is quick (half life of ca. 30 min) it would be challenging to be observed separately in aquatic systems and thus will be seen as part of the overall UV-light-driven reaction. We hoped that we could convey this 2-stage process clearer now.
Thank you for the proposal of a catalyzed reaction by transition metals. We picked it up in the text.

Changes in the text:

*… Accordingly, we see that a ROS-induced production pathway has the potential to explain excess alkanes in our samples, however, little is known about ROS chemistry in ice cores in particular for reactions with organic precursors and more research is needed to understand the role of ROS in organic decomposition in ice. Another alternative to the two-stage reaction pathway with ROS would be a reaction catalyzed in the meltwater by dust-derived transition metals. This has been observed for example for the oxidation of $SO_2$ in water-activated aerosol particles (Harris et al., 2013), but to our knowledge has not been described in the literature for alkane production via organic precursors so far. Accordingly, we can only speculate on this pathway at the moment.*

**Line by line comments in the order that appear in the manuscript:**
Ln. 26: not "…from CH4 excess" but 14 to 91 ppb of CH4 excess.
Done/ changed

Ln. 33-34: Simplify the sentence to report "carbon isotopic signature of excess methane is … and its deuterium isotpic signature is…".
Done/ changed

Ln. 32-39: Should be revised based on detailed comments above.
Done/ we changed the text, so that it becomes more clear that we do not definitely rule out a thermogenic source.

Ln. 42-45: These sentences can be deleted; they are better placed in the conclusions.
Done/ deleted

Ln 52-55: As it is written, this sentence applies to all trace gases mentioned in the previous sentence, but it really only applies to CO2, CH4, and N2O. You do not have to mention ethane and propane in the preceding sentence, I don't think.
Yes indeed, ethane and propane are not really contributing to global warming and we deleted "and the global warming caused by it".

Ln. 72: Ice cores from Greenland and Antarctica instead of bipolar. Or explain that bipolar means this in the parenthesis.
Done/ changed

Ln. 93 and ln. 98: Records instead of record.
Done/ changed

Ln. 100-106: It may be beneficial to reproduce a figure from Lee at al. (2020) and show in the Appendix; it is difficult to follow this without simultaneously looking at figures from Lee et al. (2020).
Lee at al. (2020) is published in *Geochimica et Cosmochimica Acta*, we therefore refrain from reproducing figures to avoid copywrite infringements.
And instead of adding an extra figure to follow our description, we realized that the depth of detail might not be necessary for the reader here, so we shortened this paragraph and deleted unnecessary information.

Ln. 108-109: Replace "- as well measured with … -" with ", which was also measured with…,".
Done/ changed

Ln. 127: No need for "With their data" in the beginning of the sentence.
Done/deleted

Ln. 171: Best if you provide the ages at the depths, perhaps in parathesis.
Done

Ln. 171: The NGRIP samples are… instead of "stem".
Done/ changed

Ln. 175: It is not clear what quantities were measured in 2011 and 2018.
In 2011 we measured $\delta^{13}$C-CH$_4$, ethane, propane, only with the 1$^{st}$ extraction.
In 2018 we measured methane, ethane, and propane, $\delta^{13}$C-CH$_4$, $\delta$D-CH$_4$, 1$^{st}$ and 2$^{nd}$ extraction
No changes in the text.

Ln. 180: What age/depth range are the EDC samples from?
They are from MIS 4.
Added to the text.

Ln. 199: It is not clear what the selection criteria are. Are you looking for high dust, stable methane, for example?
The explanation of the selection criteria can be found just in the paragraph above.
No changes in the text.

Ln. 205: Leading to higher Ca/dust ratio or a variable Ca/dust ratio given what you say in the beginning of the sentence?
Done/ changed

Ln. 206: You don't need to say "on the NGRIP depth" in the sentence.
Done/ deleted

Ln. 210: You should briefly explain why you are not using standard linear regression.
Done/ changed in the text.
See also comment further below.

Fig. 1: It is difficult to tell green lines from cyan lines.
Cyan was replaced by turquoise in the text.
I checked all color combination in all figures with the colorblindness simulator.

You should provide a brief explanation of why you are showing the AICC2012 gas chronology; this can be very confusing for non-specialists. You can consider presenting actual plots vs. time in the appendix instead of showing non-linear age scales as top axis.
For NGRIP, the AICC2012 gas scale is the only consistent gas age scale and public available. We think it is most robust to plot all parameters (gas and ice parameters) on the depth scale as we

are talking about effects on the gas composition that are dependent on the ice composition at the same depth. This prevents confusion as otherwise we would need to plot the gas phase parameter $CH_4$ on an ice age scale or $Ca^{2+}$ on the gas age scale. In order to give the reader some age information the respective gas age or ice age we provide the two age scales on the top of the figure.
No changes in the text.

I notice the delta-age exceeds 1000 years at times, is this to be expected at NGRIP?
Yes, for the times with the lowest accumulation rate.

Also, best to state the extraction method of the methane record in the caption.
Done

Ln. 230-231: You only need to say the data for other NGRIP bags are shown in appendix A.
Done

Ln. 275-276: State the assumptions for the ppb blank calculation.
Done/ see also comments above

Ln. 283-288: 100 min at 0degC is quite long. Does the water freeze again at some point, or is it above 0degC?
It stays above 0°C / does not freeze again during the procedure.

Changes in the text:
*... After melting is completed, the temperature of the melt water is stabilized close to 0°C, but does not refreeze again.*

From Fig. 3, it looks like the vessel is not exposed to the activated carbon trap during the 100 min?
Yes, during the 100 min waiting time the vessel is closed and not connected to the carbon trap. This leads to an accumulation of the produced alkanes as visualized with the green dots in the water.

Changes in the text:
*... After all sample air is collected in the 1ˢᵗ extraction, the meltwater is left in the isolated sample vessel (the vessel is closed and not connected to the carbon trap) and held at temperatures close to 0°C for ~100 min (step d).*

The actual sample spends 35 min melting, then 14 min of He sparge if I understand correctly.
Yes, right!

You state 24 min sparge for the 2nd extraction, but Fig. 3 shows 24+14 min?
We clarified this in the text.

Changes in the text:
*... After this "waiting time" of ~100 min, He is purged through the meltwater for ~24 min to extract the gases that have been accumulated during this time interval simulating the extraction time of the 1ˢᵗ extraction, followed by another ~14 min of He purging to mimic the last step of the ice extraction when the sample had completely melted (step f).*

How much He do you typically use to sparge how much water?

4 mL min$^{-1}$ at STP through the meltwater of a ∼ 150g ice sample. Note, the actual volumetric flow of the He flow at the pressure present within the vessel through is much higher as the pressure in the headspace above the meltwater is in the order of 10 to 50 mbar (thus the volumetric flow is about 100 - 400 mL min-1 at the pressure of the vessel headspace).

Changes in the text:
*... After melting is completed, the temperature of the meltwater is stabilized close to 0°C, but does not refreeze again. Afterwards, He is sparged with 4 mL/min at standard temperature and pressure (equivalent to 100-400 mL at the varying low pressure in the headspace) through the melt water for ~14 min through a capillary at the bottom of the vessel to transfer any remnant gas species dissolved in the melt water onto the AirTrap (step c).*

Ln. 293: Did you conduct two extractions with the EDC/Talos Dome samples and use the 2nd extraction as the blank?
Yes, for those EDC samples we conducted a 2$^{nd}$ extraction to mimic the conditions we have for the NGRIP samples. Since the Antarctic ice core samples do not show any visible *in extractu* production, we regard these 2$^{nd}$ extractions of EDC samples the most realistic quantification of our blank contribution.

See comments above and changes in the text about the blank determination.

Ln. 299-300: Are these the 2nd extraction moles converted to blank ppbs after correction with the system blank as determined by the EDC samples?
The blanks are not subtracted from the measured values in the 2$^{nd}$ extraction as stated in the main text.

Ln. 320-321: You do not need the second sentence of the paragraph.
We kept it, and added the typical sample size (see next comment) for consistency to the analogue section above.

Ln. 323: How many grams of ice is typically used for this analysis?
∼ 300g (added to the text).

Ln. 331: Do you mean "Same as the 13CH4 method.."?
Yes/ Done

Ln. 336-337: I'm confused about these times. In the beginning of the paragraph, you say 25-30 min melt plus ~40 min sparge, but then the total is ~90 min? And for 13CH4, you previously stated 35 min plus 14 min of H He sparge, here you are stating 35 min as the total amount of time from the appearance of melt water to the completion of the He sparge. Does it take about 14 min to start melting the samples then?
Yes / we clarified this in the text

Ln. 355-356: This sentence is constructed as a conclusion, but we do not know this for ethane and propane; you did not even show the data yet. You can say you expect that the

in extractu component might dominate the signal for ethane and propane, for example.
Done/changed

Ln. 386-387: It worth mentioning here again that it is the unusually high mixing ratios that leads you to believe ethane and propane is glacial ice do not represent atmospheric levels. Otherwise, it is not unexpected that ethane and propane would change proportionally in the atmosphere.
Done/changed

Ln. 387-394: Describe how the errors are calculated in the appendix or as supplement. You are talking about error propagation, but I cannot understand which errors are being propagated.
We now clarified in the text that we weighted both the average and the standard deviation by the number of samples. We feel like this is sufficiently explained now.

Fig. 4: The data point designated as an outlier is not really an outlier among the four data points from that one bag. I'm not convinced it is an outlier even when considering all data from all bags. I doubt that not including this one pint in your calculation of the average changes anything. No need for this.
You are right, the term "outlier" is misleading here as it is not an outlier in a statistical sense. Instead, we have called it "flagged sample" as we are not sure if we can 100% trust this measurement as one vent (V6) was not closed during the measurement.

Changes in the text:
*... Note that there is a flagged sample for $CH_4$ in bag 3453 (yellow asterisk in Fig. 4), where one vent (V6) was unintentionally open during the measurement, which may have compromised the result. We therefore excluded the production ratio determined from bag 3453.*

Also, report what the slope of this figure is, then you can also show the inverse to report methane/ethane ratio.
The methane/ethane ratio is reported.
No changes in the text.

Fig. 5: When talking about the high Ca data from bag 3515, there is higher uncertainty in the Ca estimate for that data point; that is all you can say. You cannot claim that there is likely a bias of few hundred ng/g.
Done/ deleted

Ln. 402-403: Refer to Fig. 4 at the end of the sentence.
Done

Ln. 404-406: Would be better if you can offer an estimate of the spectral width of the firn smoothing and compare with the spans of gas ages for your bags.
Unfortunately, we don't have the exact information for NGRIP for the spectral width. A back of the envelope calculation (extent of lock-in zone in cm WE/ accumulation in cm WE) provides a rough estimate of the width of the gas distribution on the order of 50-100 years.
No changes in the text.

Ln. 410: Instead of "which is calculated from…", better to say "which can be estimated from…".
Done/changed

Ln. 412-425: Does the magnitude of the xsCH4 matter for this paper? If you want to claim that using ethane is a better predictor of xsCH4, then you should show an analysis of Ca vs CH4 and ethane vs methane together, perhaps in Fig. 4 as a second panel, and also factor in the uncertainty that arises from assuming 0.39 ppb of atmospheric ethane.
No, it does not matter that much. We only state that using ethane is a better predictor of $CH_{4(xs)}$ than $Ca^{2+}$ as the correlation is higher.
0.39 ppb is the best estimate for the Holocene but it is likely lower for glacial times. However, analyzing the uncertainty arising from this estimate is beyond the scope of this study and not relevant for the discussion of our results.
No changes in the text.

Ln. 436-459: There is a lot of repetitive information here. For determining a "best predictor" for xsCH4, see my comment above.

Ln. 461-464: This has been said before.
Done/ deleted

Ln. 491-493: After the comma, all you have to say "assuming that all alkanes in the 2nd extraction were produced…"
Done/ deleted

Ln. 498-506: Show the 1st vs 2nd extraction data for methane and propane in the appendix. I find it confusing that the methane ratio is very similar to ethane and propane when we are assuming most of the measured methane in the first extraction is atmospheric in origin. Are you plotting about xsCH4 estimate from 1stextraction vs. the 2nd extraction methane?
There must be a misunderstanding here. No atmospheric part is included in the ratios – all ratios are based on excess alkanes.
Regarding the overall number of figures, we decided to show only the results for ethane between the 1st and 2nd extraction as the plot for propane is very similar. For methane and propane, this information is given in the text, but not shown in a separate figure.
No changes in the text.
Ln. 507-508: The first sentence of the paragraph is not needed.
Done/ deleted

Ln. 519: In the example shown for ethane, …
Done

L 513-525 and Fig. B1: Explain what you use as the time axis data for Fig. B1. The figure captions mention a Monte Carlo approach, it is not clear what has been done here. If you show ln[A] vs. time, it would be easier to evaluate the validity of 1st order process starting from zero ethane. For all samples, 3rd extraction is above the model while the 2nd is below. Are you assuming zero ethane at t=0? Yes
It is difficult to evaluate from 3 data points, but this look like even a slower process than what your half-life estimates suggest, with excess ethane likely to continue coming out after the 3rd extraction.

We realized that the equations and assumptions used for Figure C1 were too condensed and not clear enough. We now provide the underlying equations and more background information as part of the appendix (see below the new text and the updated figure).
With regard to your suggestion to plot ln[A]: We thought about that, but this more elegant approach is mathematically not possible in our case as we have only the cumulative amount of the product rather than a measure of the declining amount of the precursor.
And you are right, our exponential decay model might not capture a more prolonged small production. The key message of this kind of analysis is that the alkanes are clearly not produced in a constant way but production within the first 30 min is dominant while the production during the next hours is rapidly dropping. This is now clearly illustrated in the appendix.

Note that we have also numbered the 5 GRIP samples in Fig. 7a, in order to be able to relate the individual samples to Fig. C1.

Changes in the text:

[Figure]

*Figure C1: **Temporal dynamics of excess ethane production in GRIP ice core samples.** Cumulative ethane amount from the 1st, 2nd, and 3rd extraction in relation to the time available for a potential reaction in the melt water during each extraction. We assume a first-order reaction kinetic as model for our observations where the mean half-life time (τ) and standard deviations are calculated for each GRIP sample from the compilation of all 1000 iterations of our Monte Carlo approach. The numbered samples can also be found in Fig. 7a.*

**These equations below will be accompanying the Fig. C1 and part of the Appendix C.**

*The general equation to describe a first-order chemical reaction or exponential decay process (e.g. release of adsorbed gas from the adsorbent) is Eqn. (1)*

$$N(t) = N_0 * e^{(-t/\tau)} \qquad\qquad (1)$$

*With $N_0$ the total amount of substance (reactant) at the start of the reaction. $N(t)$ equals the remaining amount of the reactant at time t, and t being time of reaction and $\tau$, the mean lifetime of the reaction. In our case, we cannot determine $N(t)$ neither do we know $N_0$ but we experimentally determined the cumulative amount of the product, $P_{cum(t)}$, at three different times as our observable quantity. Thus, in Eqn 2 we define $P_{cum(t)}$ as the difference between $N_0$ and $N(t)$.*

$$P_{cum(t)} = N_0 - N(t) \qquad\qquad (2)$$

*Replacing $N(t)$ in Eqn (1) with our definition in Eqn (2) we obtain Eqn (3), which contain two fit parameters, $N_0$ and $\tau$, as well as our observable parameter $P_{cum(t)}$, i.e. the cumulative amount of alkane for a certain time step.*

$$P_{cum(t)} = N_0 - N_0 * e(-t/\tau) \qquad\qquad (3)$$

*For the 5 GRIP samples we have three consecutive measurements each, the 1st, 2nd, and 3rd extraction. The time dependent $P_{cum(t)}$ values are as follows: $P_{cum0}$ is defined as 0, representing the state of the unmelted ice sample before liquid water is present. $P_{cum1}$ is the measured amount from the 1st extraction (ice extraction) minus the estimated contribution from the atmosphere and minus the blank contribution for the 1st extraction. $P_{cum2}$ is the sum of $P_{cum1}$ and the value from the 2nd extraction minus the blank contribution of the 2nd extraction. Similarly, $P_{cum3}$ is the sum of $P_{cum2}$ and the value from the 3rd extraction minus the blank for the 3rd extraction.*

*To account for the uncertainties of the involved measurements and corrections, we added normally distributed errors to the following parameters (measured value $\pm$ 5 %; blank $\pm$ 20 %; atmospheric contribution $\pm$ 50 %), we also assigned an uncertainty of 5 min to the time to account for variations of the melting speed of the ice and delays between the individual measurements (1st, 2nd, 3rd).*

*For the fitting procedure we used the Matlab built in nonlinear least-squares solver called 'lsqcurvefit' and performed 1000 runs where we varied the above-mentioned input parameters. The output of the function are the two fit parameters, i.e., $N_0$ and $\tau$. From the 1000 runs we calculated the mean and the 1 sigma standard deviation of the lifetime.*

*Note, this approach can only be suitably applied to ethane and propane as the past atmospheric contribution for these gases in the 1st extraction is typically small against the excess contribution for dust-rich samples. For our 5 GRIP samples, where we have three consecutive extractions, 4 samples are considered "dust-rich" and are suitable to provide robust estimates for $\tau$. In contrast, one sample is from an interstadial period with very low dust content and thus shows negligible production of alkanes in all three extractions. While this sample is not suited to provide robust estimates for $\tau$, this sample allows to assess the first-order plausibility of the blank correction and the assumed atmospheric background for ethane for the 1st extraction (sample number 5, bottom-most sample). For a sample without any in extractu production, the cumulative curve should be flat at around 0 which is the case within our error estimates.*

Ln. 520-521: This observation would hypothetically be mimicked by exponential decay of a precursor organic, if one exists.
We have re-phrased this sentence.

Ln. 529: I only see ethane in Fig. 7b.
Done/ changed

Ln. 545-556: It is possible that excess alkanes are a result of more than one process. For example, the 1$^{st}$ extraction can include an in situ component, no?

No. For methane we have the evidence from the CFA measurements which do not show any imprint of dust/ Ca$^{2+}$ on the measured CH$_4$ concentration. Thus, for CH$_4$ it is settled that there is no in situ component with dust. However, for ethane and propane we don't have yet presented this proof but derived a common pathway for both the 1$^{st}$ and the 2$^{nd}$ extraction from the similar production ratios. Meanwhile, we did experiments with our new sublimation device (Mächler et al. 2023, AMT) where we sublimated dust-rich NGRIP ice and transferred the released air to our $\delta^{13}$C-CH$_4$ device. In short, the result was that methane, ethane and propane are not elevated for these samples, ruling out an in situ component and indicating that the 3 alkanes are not produced or released during vacuum conditions. Currently we do not mention this result in our paper as it would make the paper longer and we felt that it is not a key information that would change the interpretation of our data. If needed, we could add a few sentences on these results if you wish, but otherwise we leave it out.
 No changes in the text.

I do not understand what you mean by "…only a proxy for higher in extractu production" on line 556.

With this sentence, we want to say that Ca$^{2+}$ or mineral dust is not the precursor itself, but just a proxy because of the high correlation. We clarified this in the main text.

Changes in the text:
*… We propose that this reactant co-varies with Ca$^{2+}$ and particulate dust, where Ca$^{2+}$ is of course not a reactant itself and represents only a proxy for higher in extractu production.*

Fig. 8a: Something wrong with the top x-axis scale?

I changed the tick size

Ln. 663: Did you conduct isolation test over frozen ice?

Yes, this is done by default in every $\delta^{13}$C-CH$_4$ measurement for this measurement series.
A blank over the ice or "He over the ice sample" processes the same amount of He and the same trapping times and subsequent steps during this blank measurement as for an ice sample measurement, while the ice sample is present in the cooled vessel at -5°C. Details can be found in Schmitt et al. (2014).
No changes in the text.

Ln. 691-693: Better to say deemed unlikely than ruled out.

Done/ changed

Ln. 693-697: Not necessary. You come back to all this later and say the same things again.

Done/ deleted

Ln. 735: Fractionation can also happen desorption.

Done/ changed

Ln. 737-740: Insufficient for what?

"insufficient" might not be a good word here. I changed it to "minor"

What desorbs could be replaced by what is present in the environment. I don't understand how you rule out desorption in the firn either.
We clarified this in the text.

Ln. 742: The source of dust to Greenland has been mentioned multiple times. Bets to say it once when you are ready to offer discussion about why the source region is relevant for this paper.
Done/ deleted

Ln. 747-748: I suggest rephrasing to "clay minerals have high adsorption capacity and retention potential for alkanes" if this is what you mean.
Done/ changed

Ln. 750-753: This information is not relevant for this paper unless you are going to report the different types of clay mineral found in glacial Greenland ice.
Done/ deleted

Ln. 785: Is comparable adsorption for all alkanes a reasonable assumption?
We also question if methane, ethane, and propane are equally adsorbed on dust particles. However, this cannot be verified.
If a different adsorption characteristic exists for the different alkanes, e.g. due to weight/size of the molecules, this would misrepresent the original ratio of the source.
We have clarified this in the text.

Changes in the text:
*… To explain the constant ratio of methane, ethane, and propane of 14:2:1 in our samples with an adsorption mechanism, we need to discuss the potential origins of the adsorbed alkanes. First, we find very high relative excess contributions of ethane and propane in our samples, while we see a small excess contribution for methane compared to the atmospheric background. If we assume a comparable adsorption for all three alkanes, this would imply a strong relative enrichment of ethane and propane over methane in the concentration of these gases during adsorption. This is not in line with the past atmospheric $CH_4/(C_2H_6+C_3H_8)$ ratio where past atmospheric ethane concentrations by Nicewonger et al. (2016) are an order of magnitude smaller (and propane concentrations even less) than the measured concentrations in our NGRIP and GRIP ice core samples.*
*In contrast, the ratio of methane, ethane, and propane for our samples of approximately 14:2:1, translates into a $CH_4/(C_2H_6+C_3H_8)$ ratio of ~5, which is most consistent with a thermogenic origin (see Fig. 11, left panel). However, due to the different adsorption capacity of mineral dust particles, also a fractionation of the three alkanes is to be expected during the adsorption process, which could alter the thermogenic signature.*

*… For the seep adsorption scenario A1 to work the dust particles on which the thermogenic gas adsorbed are now not allowed to experience any contact with liquid water prior to the analysis in the lab. In other words, if the particles get in contact with liquid water after the adsorption step, the adsorbed alkanes would desorb from the particles as they do it in the laboratory during melting. Given the occurrence of wet/dry cycles in the source area (Ruth et al., 2007), we question the plausibility of scenario A1. Moreover, we expect the characteristic desorption time to differ between the three alkanes, which would be in contradiction to the*

*observation that the alkane ratios in the 1st and 2nd extraction are the same within the error limits.*

*… The second part of a potential M1 process, the adsorption of the microbially produced excess alkanes onto dust particles in the ice and the subsequent desorption during extraction, remains difficult to assess. A selective adsorption of the in situ produced alkanes on mineral dust in the ice requires that the in situ production is taking place on the dust particles themselves, which can be questioned but cannot be ruled out. However, our ratios of excess methane/ethane/propane in NGRIP and GRIP samples add another piece of corroborating evidence that excess alkanes are not produced microbially. The main microbial production process of methane, the decomposition of organic precursors in an anaerobic environment by archaea, also co-produces ethane and propane, however only in marginal amounts. The typical methanogenesis yields >200 times more methane than ethane and propane (Bernard et al., 1977; Milkov and Etiope, 2018) while we find a molar ratio of methane to ethane to propane of 14:2:1 in our samples. This renders a microbial production pathway (in situ and in extractu, i.e. M1 and M2) for excess alkanes unlikely. Moreover, a microbial production of CH$_4$ is unlikely in view of the $\delta^{13}C$-CH$_{4(xs)}$ signature which is too heavy for microbial CH$_4$.*

*Similar to our argument made for the pure desorption hypothesis, the constant excess alkane ratio in the second and first extraction is difficult to reconcile with an expected different desorption lifetime for the three alkanes.*

*… Moreover (as mentioned before), in view the expected different desorption characteristics of the three alkanes we would expect different alkane rations in the 1st and 2nd extraction, which is not the case. Accordingly, a direct abiotic production during the melt process appears to be more likely than a desorption process.*

Ln. 793-796: Why lower limit?
Done/ deleted

Thermogenic ratios cover a very large range in Fig. 11. The star is in orange, but pink is actually the pure thermogenic signature, no? Don't we have to consider the kinetics of desorption in these discussions as well? Is that likely to change for smaller/larger or lighter/heavier molecules?
Yes indeed, we would expect that methane desorbs quicker than ethane or propane. But we do not observe this as the ratios between the 1$^{st}$ and the 2$^{nd}$ extraction are very similar.
No changes in the text.

Ln. 804: Don't you have to use the atmospheric values at the respective ice ages of your Greenland samples for this discussion?
The measured values for Greenland samples are affected by *in extractu* production. While the $\delta^{13}$C-CH$_{4(xs)}$ signature is close to atmospheric values, the $\delta$D-CH$_{4(xs)}$ signature of about -300 permil is much lighter than the atmospheric values. Hence the difference between $\delta$D-CH$_{4(xs)}$ (about -300 permil) and atmospheric $\delta$D-CH$_{4(xs)}$ values (-80 permil) is 220 permil! As a first approximation we can take the NGRIP $\delta$D values that contain only little dust (i.e. our own values from Fig. 9) and low *in extractu* contribution, and therefore very close to atmospheric values. We deleted unnecessary information.

Ln. 809-810: Lower meaning they are lighter? 1permil different in which direction?

See comment above

Ln. 823: Your delta age is on the order of 500-1000 years during MIS2 and 3 based on your figures. If methane levels and isotopic signature is changing in the atmosphere, the adsorbed methane can alter the atmospheric signal preserved in the ice even if the adsorbed methane is atmospheric in origin.
Indeed, if the atmospheric $\delta D$-$CH_4$ values changes drastically on time scales of the delta age (500 to 1000 years) then this slightly older adsorbed atmospheric $CH_4$ (ice age) would bias the younger (gas age). Yet, the observed $\delta D$-$CH_4$ changes in Antarctic and Greenland ice cores are only on the order of 10 to 20 permil, thus the leverage is tiny. Therefore we say "would not be able to strongly affect".
No changes in the text.

Ln. 835: It seems to me that the deuterium isotopic signature is also within the thermogenic range given Fig. 11, especially considering the Lee et al. estimate of the deuterium end member.
Done/ clarified in the text

Ln. 852: Replace "regards" with is.
Done

Ln. 853-861: Microbial production can happen on the dust particles, or even in the pores of the dust particles, no?
We ruled out a microbial origin anyway based on the carbon isotopic signature. Additionally, the typical size of bacteria (0.5 to 5 microns) is similar to the typical size of the dust particles.
No changes in the text.

Ln. 863-891: Is this not a discussion of what you say at the end of the previous paragraph that you will not discuss?
Well, yes and no. This paragraph describes the 1st part happening in a potential M1 process: the microbial production. The 2nd part of this process would then be the ad-/desorption.
The 1st part only (without an ad-/desorption process) is ruled out and will not be further discussed.
But I see your point. The paragraph is also not very relevant for the M1 process, so we decided to delete it.

Ln. 893-894: Microbial activity could presumably happen on/within the dust particles. I would expect microbes to be transported to the ice sheet on aerosols anyway, or do you think that they are independently transported?
This sentence is about the "microbially produced excess alkanes" but not the microbes itself. But we ruled out microbial production anyway.
No changes in the text.

Ln. 896-902: I agree that the poison experiments of Lee et al. rule out microbial production during the extraction, but not if the microbial production happened before hand and you are merely seeing that stuff desorb until low pressure and in the presence of water during the extraction. I do agree that the alkane ratios are a good argument to rule out microbial production.

The microbial production cannot have happened before, please see sections with the CFA evidence.
No changes in the text.

Ln. 924-928: A purely chemical reaction is also more likely to happen on the dust surface itself, and more likely when metal particles that can facilitate reactions are present.
See also answer to comment on line 556
We think, that mineral dust or $Ca^{2+}$ is just a carrier for any labile organic substances and the dust particle itself is „passive" in the reaction. At the moment, there is no experiment to show where exactly the reaction happens. This could be an approach for further methodological analyses/ experiments.
No changes in the text.

Ln. 940: Do you mean you consider this pathway plausible?
Yes/ changed

Ln. 956-961: These two sentences are not needed.
Done/ deleted

Ln. 963-990: Any of the studies show ethane/propane production?
Yes: McLeod et al., 2008 /John and Curtis, 1977 / Dumelin and Tappel, 1977 / Derendorp et al., 2010, 2011
No changes in the text.

Ln. 1035-1036: How about different materials used in different extraction systems?
What do you mean with "materials"? In the system itself, e.g. the sample vessel or traps? Sorry, we have no statistics on that.
But the sentence in line with Ln. 1035-1036 "Thus, any excess $CH_4$ in measurements from different labs performed under different conditions may differ" is about different "conditions", e.g. temperature of the meltwater, extraction time, … and this can definitely play a big role regarding the experimental results.
No changes in the text.

---

## Author Comment (AC2)

This study provides a systematic investigation into the origin of "excess" methane measured in dusty [Greenland] ice core samples. The authors build on previous work of Lee et al., 2020, adding d13C-CH4 and dD-CH4 isotopic data as well as evidence of coproduction of ethane and propane. Overall, this work comprises an important contribution to the literature by advancing our understanding of the potential mechanism(s) responsible. Congratulations to the authors on an excellent set of measurements. I have a few suggestions, mostly minor or even grammatical. I would like to highlight the need to tighten up the discussion of uncertainties (marked * below), particularly in light of Reviewer 2's misgivings.

**Comments listed in line order:**

Title: Suggest removing first 'excess'.
Done/ deleted

L23: kyears should be kyr or ka
This is an editorial requirement

L26 and throughout: Why is 'in extractu' italicised and 'in situ' is not?
This is an editorial requirement

L26-29: Can the term 'excess' be defined here when first used? It may not be obvious to many readers. It is defined at L121 after being used several times.
The term is only used before in the abstract, however, no definitions/explanations should be used here. The first presence is in line 121, where it is then explained.
No changes in the text.

L31-32 and throughout: Can a threshold for 'dusty' ice be defined up-front in the abstract? If the excess alkanes scale with dust content then where to you draw the line? When is dust content low enough for this effect to not be a problem/not detectable?
Based on our current knowledge, the amount of excess alkanes scales linearly with the amount of mineral dust within the ice samples. Therefore, indicating a threshold would be misleading. But for a better understanding we re-structured the text accordingly.

Changes in the text:

*Abstract.* *Air trapped in polar ice provides unique records of the past atmospheric composition ranging from key greenhouse gases such as methane (CH₄) to short-lived trace gases like ethane (C₂H₆) and propane (C₃H₈). Recently, the comparison of CH₄ records obtained using different extraction methods revealed disagreements in the CH₄ concentration for the last glacial in Greenland ice. Elevated methane levels were detected in dust-rich ice core sections measured discretely pointing to a process sensitive to the melt extraction technique. To shed light on the underlying mechanism, we performed targeted experiments and analyzed samples for methane and the short-chain alkanes ethane and propane covering the time interval from 12 to 42 kyears. Here, we report our findings of these elevated alkane concentrations, which scale linearly with the amount of mineral dust within the ice samples. The alkane production happens during the melt extraction step of the classic wet extraction technique and reaches 14 to 91 ppb of CH₄ excess in dusty ice samples. We document for the first time a co-production of excess methane, ethane, and propane with the observed concentrations for ethane and propane exceeding their past atmospheric background at least by a factor of 10. Independent of the produced amounts, excess alkanes were produced in a fixed molar ratio of approximately*

*14:2:1, indicating a shared origin. The carbon isotopic signature of excess methane is (-47.0 ± 2.9) ‰ and its deuterium isotopic signature is (-326 ± 57) ‰ in the samples analyzed. With the co-production ratios of excess alkanes and the isotopic composition of excess methane we established a fingerprint that allows us to constrain potential formation processes. This fingerprint is not in line with a microbial origin. Moreover, an adsorption-desorption process of thermogenic gas on dust particles transported to Greenland appears not very likely. Rather the alkane pattern appears to be indicative of abiotic decomposition of organic matter as found in soils and plant leaves.*

L36: change 'confine' to 'refine'
Done/ changed

L67: "the good guy" – can a less gendered term be used here? Maybe methane is female…
Done/ changed

L74: should be "relative" contribution?
Done/ changed

L108: change 'as well' to 'also'
Done/ changed

L115-116: I couldn't see this thesis available online…could the magnitude of the Antarctic dD variations at least be quantified here please?
No, unfortunately this is not available online. We therefore changed it to "unpublished data", and we also give a value (3-4 ‰).

L159: 'evidences' should be 'evidence'. 'hypotheses PROPOSED by Lee'
Done/ changed

*L272 (then also L295, L339 etc.): All mention of precision or uncertainty needs to be clarified. Are you talking about a 2 sigma precision here? Are these values obtained from repeated/pooled measurements? L443 mentions a 2 sigma uncertainty.
Sorry for the confusion and thank you for this important notice! The uncertainties given in the numbers are 1 sigma here and throughout the manuscript. We changed it accordingly. This implies, that the results are not significantly different within the 2 sigma error, as stated in our manuscript.
Reading the comments from both reviews, we also realized that to prevent ambiguities we need to better explain how we dealt with the blank contribution throughout the manuscript. Throughout the paper, we showed the data without blank correction, i.e. we plotted the actual values of the measurements which is the ice core derived amount plus the amount derived from the system (blank). Except for the $CH_4$ amount measured in the 2nd extraction which has a considerable amount of "blank" contribution, the system blank values for ethane and propane for both the 1st and the 2nd extraction are sufficiently small compared to the sample-derived amount. The advantage of showing the non blank-corrected values is that we can plot both the blank and ice core measurements in a single figure which allows to see the size of the blank contribution and also the respective alkane ratio of the blank contribution.
With regard to $\delta^{13}C$-$CH_4$, indeed the $\delta^{13}C$-$CH_4$ signature of the blank (EDC) is similar and only a few ‰ heavier (-39.0 ‰) to the signature of our Greenland samples. Applying an isotope mass balance approach, we see that the leverage on our NGRIP values is small (0.31 ‰). Thus,

applying a blank correction has only little leverage but would shift the sample values a bit towards isotopically lighter values, and therefore more into the direction of the values obtained from the Keeling plot approach. For the sake of the length of the paper, we did not expand on these corrections for the $\delta^{13}$C-CH$_4$-signature. We wish to stress that measuring the $\delta^{13}$C-CH$_4$ signature of such small CH$_4$ samples as available from the 2$^{nd}$ extraction is at the edge of what is possible with our measurement device.

We clarified in the main text that we are not performing a blank correction (except for Fig. C1 where the blank correction is necessary).

We also clarified the section explaining the blank determination and precision of our method.

We added another Figure (Appendix B, Figure B1) with a description of the blank determination.

Figure 8 is revised.

[Figure]

Changes in the text:

[revised manuscript text omitted]

Figure 4: grey hatched area doesn't show up.
The grey hatched area is very small, only 0-0.39 ppb of ethane. It is the background concentration and just given to illustrate the difference to the measured values. It should therefore not stick out, but of course it should be visible. I strengthened the shade of grey.

L491: Sorry I don't get the meaning here…how do you define the gas extraction as quantitative (or qualitative)?
Done/ deleted (as also desired by reviewer No.1)

*L581 onwards and Figure 8: Looking at the figure, the intercepts do not overlap within uncertainties given, yet the text suggests they do…please clarify.
*L601: as above
Sorry, it is again 1 sigma here.

L695-697: Suggest removing this last sentence – it is confusing and implies you actually know something concrete about the rate of desorption.
Done/ deleted

L727: Could 'deflation' be defined, for those not familiar?
Done

Figure 10: Great figure.

Thank you.

L743: Please check papers cited here. Bory et al., 2003 did not analyse glacial dust samples, Rhodes et al., 2013 did not analyse any dust.
Done

L747: Doesn't make sense as written.
Done/ clarified in the text

L935: phrase not sentence
Done/ clarified in the text

L937: Could you expand – what is abiotic conditioning?

Changes in the text:
*… We stress that although we can exclude a direct UV effect during sample extraction, it is possible that UV irradiation during dust aerosol transport to Greenland and within the upper snow layer after deposition until the snow gets buried into deeper layers may precondition organic precursors attached to mineral dust to allow for alkane production to occur during extraction. In particular, the first step of the reaction (excitation of the homolytic bond of a precursor compound) may start already in the atmosphere or in the upper firn layer where energy from UV radiation is available. Within the ice sheet the reaction may be paused ("frozen reaction") and only becomes reactivated during the melting process when liquid water is present.*

L1011: Seems difficult to reconcile the dust coming from desert regions and it being rich in organic material.
Dust from the Taklamakan (and Gobi) desert might not be "rich" in organic material, but it definitely contains organic material, and it can accumulate organic matter during transport and organic aerosol formation in the atmosphere.
Ventura, A.; Simões, E.F.C.; Almeida, A.S.; Martins, R.; Duarte, A.C.; Loureiro, S.; Duarte, R.M.B.O. Deposition of Aerosols onto Upper Ocean and Their Impacts on Marine Biota. *Atmosphere* 2021, *12*, 684. https://doi.org/10.3390/atmos12060684

Huo, W., He, Q., Yang, F. *et al.* Observed particle sizes and fluxes of Aeolian sediment in the near surface layer during sand-dust storms in the Taklamakan Desert. *Theor Appl Climatol* 130, 735–746 (2017). https://doi.org/10.1007/s00704-016-1917-4

The uptake of organic substances during transport is explained in the text:
*… Organic precursors for this abiotic production during extraction could be any organic matter (either microbial or plant-derived). As the amount of excess alkanes is tightly coupled to the amount of dust, we assume that these organic compounds are attached to dust particles. This "docking" of the organic precursor onto the mineral dust could happen already in the dust source region involving organic material available at the surface. Or it could happen by adhering of volatile organic molecules or secondary organic aerosols from the atmosphere to the mineral dust aerosol either before deflation at the source region or during transport to Greenland.*
No changes in the text.

L1036: Can you go further? Could this explain previously reported lab offsets?
Unfortunately not.
Previously reported lab offsets concern Antarctic ice samples, which do not show any signs of *in extractu* production. Explaining lab offsets for Greenland ice samples would be beyond the scope of this paper.
No changes in the text.

L1158: Should 'contradicting' be 'corroborating'? The meaning is not clear here.
No, "contradicting" is right. What we want to say here, is, unless we do not have anything specific speaking AGAINST the ROS hypothesis, we see this as the most likely process.
No changes in the text.

I am also in agreement with Reviewer 2 that this paper is unnecessarily lengthy, containing significant repetition (although generally well-written). This will cause many less-interested readers to give up before reaching the punchline! Some heavy-handed editing from co-authors would be beneficial. The conclusions section could be much more stream-lined with a more compelling punchline. Finally, could a summary figure or table be included which compares/evaluates the different potential production mechanisms? It is difficult to keep track through the bulk of text.
We have deleted repetitions and not ultimately necessary information.
We have also included a new table (Table 1) which displays the three hypotheses in relation to our experimental and analytical findings. This table should help to see (at a glance), which fingerprint characteristic is in line / is not in line with the respective hypotheses.

Table 1: **Overview of the different hypotheses explaining the possible sources for excess alkanes (as illustrated in Figure 10) in relation to our experimental and analytical observations.** A green checkmark indicates that the observation is in line with the respective mechanism, a purple cross indicates that the observation is in not line with the respective mechanism. A grey shaded area means that this observation does not apply or does not affect the respective mechanism.

| | (1) Adsorption-desorption of thermogenic/ atmospheric gas | | (2) Microbial production | | | (3) Abiotic/ chemical production | |
| --- | --- | --- | --- | --- | --- | --- | --- |
| | A1 | A2 | M0 | M1 | M2 | C1 | C2 |
| Correlation to $Ca^{2+}$/ mineral dust | ✓ | ✓ | ✓ | ✓ | ✓ | ✓ | ✓ |
| Alkane pattern | ✓ | ✗ | ✗ | ✗ | ✗ | (✓) | (✓) |
| CFA evidence | | | ✗ | | | | |
| $\delta^{13}C$-$CH_{4(xs)}$ | ✗ | ✓ | ✗ | ✗ | ✗ | (✓) | (✓) |
| $\delta D$-$CH_{4(xs)}$ | ✓ | ✗ | ✓ | ✓ | ✓ | (✓) | (✓) |
| $\delta D$-$CH_{4(xs)}$ estimated by Lee et al. (2020) | ✓ | ✗ | ✓ | ✓ | ✓ | (✓) | (✓) |
| Poisoning experiment by Lee et al. (2020) | | | | | ✗ | | |

---

## Author Response (AR2)

Author's response to Referee #1: Aydin, Murat

report 28 Mar 2023

Report #2

The manuscript has been substantially revised and does not need major revisions. Still, I want to clarify a few important points regarding the uncertainty estimates for the isotope data and have a couple of recommendations that I consider minor. I will state once more that measurements of dual isotopic signatures of xsCH4 is a significant advancement in the current state of knowledge on this topic.

The point I made in the first review regarding the bias in linear fits (Figs. 8 and 9) with low N was with regards to the bias in the uncertainty estimates of the fit, not bias in the fit itself. For example, the del13CH4 intercept for NGRIP bag 3453 is reported as -46.2 +- 1.5 permil (Fig. 8a). The +-1.5 permil uncertainty estimate is highly uncertain and expected to be biased regardless of the analytical regression method one uses (least squares vs. orthogonal) because N=3. I do not mind displaying the uncertainties for the individual intercept estimates on the figures simply because this is the common convention and the authors do not use these estimates during the subsequent data analyses anyway. Unfortunately, the uncertainty estimates for the weighted mean calculations are also uncertain for the same reason. That is, there are only four intercepts that are being averaged for del13C, which is akin to estimating a standard deviation from four measurements. If I'm not wrong, you use a different method for the delD data shown in Fig. 9 and calculate a weighted average of the two individual uncertainty estimates for the intercepts. This is ad hoc and somewhat unusual but OK to do in this case since there are two data points otherwise, although you should probably note this in the caption.

In any case, my main point here is that the true uncertainties in the weighted average values of del13CH4 and delD based on Figs. 8a and 9 are probably larger than the reported values. As a result, I have more confidence in the uncertainty estimate for the 2nd extraction results than the uncertainty estimates for the Keeling plots. I acknowledge that the 2nd extraction measurements are more difficult to make due to small sample sizes and the uncertainties in individual measurement are higher. However, this shortcoming is counterbalanced by the ability to determine the overall uncertainty more precisely by averaging a larger number of data points. This actually provides stronger justification to conclude that 1st and 2nd extraction results cannot be differentiated from each other with the existing data and that the delD measurements do not contradict the previous estimate by Lee et al. (2020). However, I will suggest showing the 2nd extraction results (from Fig. 8b) in Fig. 11. I would also display the estimate by Lee et al. (2020) in the same figure. I believe this would provide a more comprehensive review of all currently available information in Fig. 11. I am

comfortable with the related discussions and conclusions remaining as is, including favoring the Keeling method results for isotopic signatures in the discussions. The authors also favor the ROS explanation over the adsorption/desorption mechanism. I agree with them that the theoretical and literature-based considerations of the possible mechanisms render a chemical mechanism more likely.

There must be a misunderstanding. We use the same method to estimate the weighted average of the dD-CH4 and d13C-CH4 signature and its uncertainty. The weighted mean and its weighted standard deviation are both weighted according to the number of samples measured per bag. This is explained on the text in line 578 ff. (no changes in the text here).

We agree with the referee that the sample number is small and therefore also the uncertainty estimate has its own error. However, we stress that the y-intercepts all agree withing their calculated uncertainty. In response to the referee's comment, we added the following text to reflect this.

**Line 582-589: new text**
*.... With the small number of samples that go into the determination of the y-intercept and its error in the Keeling plot for each individual bag, the estimates of the y-intercepts and their error have to be regarded statistically uncertain. However, comparing the results for the individual bags, they all agree within each within the estimated errors. In order to get a more representative value for the isotopic signature of excess CH4 and its error, we calculate a weighted average for all bags for the y-intercept and its error. Nevertheless, this weighted error may still not be entirely representative because of the small sample number and the true error may likely be somewhat higher.*

**Line 652-655: new text**
*... As stated above, with the small number of samples that go into the determination of the y-intercept and its error in the Keeling plot for each bag, the estimates of the y-intercepts and their error have to be regarded statistically uncertain.*

**New Fig. 11a:** values for NGRIP samples from the 2nd extraction were added. The estimate by Lee et al. (2020) cannot be added to this figure, as they only estimated the dD-CH4 signature but not the d13C-CH4 value.

The uncertainties in isotopic signatures from both the 1st and 2nd extractions can be reduced by more measurements of the same nature in the future. Such work may reveal that the del13C signature from 1st and 2nd extractions are in fact different, implying a more complex situation that involves more than one production mechanism. As a final suggestion, it may be worth dropping a brief note somewhere in the text acknowledging this possibility (for example a sentence or two in section 3) so the door remains slightly ajar for unexpected results from future data sets.

Of course, our discussion and interpretation would benefit from a more certain statistics with more data points. However, we refrain from adding such a statement as you proposed for the following reasons: if we consider the possibility that more measurements could show completely different results would imply that we do not trust in our results. We refer again to the new text added (see above) which stresses that all the results agree with each other within their calculated uncertainties.

Nevertheless, we do not close the door for new insights in the future regarding the best isotopic signatures nor for our interpretation. We do not determine one mechanism responsible for excess alkanes, but we explain what we favor or consider as likely in light of our results. If there will be more (and possibly unexpected) results in the future and thus more lines of evidence, we will review the discussion about potential mechanisms again.

Line 642: Negative sign missing. It should be -341.
corrected

---

## Author Response (AR3)

Author's response to Referee

14.04.2023: no more changes to the referees' comments since the last uploaded version on 04.04.2023

Other changes in the manuscript:

- Gu et al. (2016) was changed to Qian et al. (2016)